# Opinion: Tropical cirrus — From micro-scale processes to climate-scale impacts

Blaž Gasparini[1], Sylvia C. Sullivan[2], Adam B. Sokol[3], Bernd Kärcher[4], Eric Jensen[5], and Dennis L. Hartmann[3]

[1]Department of Meteorology and Geophysics,University of Vienna, Vienna, Austria
[2]Department of Chemical and Environmental Engineering, University of Arizona, Tucson, Arizona, USA
[3]Department of Atmospheric Sciences, University of Washington, Seattle, Washington, USA
[4]Institut für Physik der Atmosphäre, DLR Oberpfaffenhofen, Wessling, Germany
[5]NOAA Chemical Sciences Laboratory, Boulder, CO, USA

**Correspondence:** Blaž Gasparini (blaz.gasparini@univie.ac.at)

**Abstract.** Tropical cirrus clouds, i.e. any type of ice cloud with tops above 400 hPa, play a critical role in the climate system and are a major source of uncertainty in our understanding of global warming. Tropical cirrus involve processes spanning a wide range of spatial and temporal scales, from ice microphysics on cloud scales to mesoscale convective organization and planetary wave dynamics. This complexity makes tropical cirrus clouds notoriously difficult to model and has left many important questions stubbornly unanswered. At the same time, their multi-scale nature makes them well-positioned to benefit from the rise of global, high-resolution simulations of Earth's atmosphere and a growing abundance of remotely sensed and in situ observations. Rapid progress on our understanding of tropical cirrus requires coordinated efforts to take advantage of these modern computational and observational abilities.

In this Opinion, we review recent progress in cirrus studies, highlight important unanswered questions, and discuss promising paths forward. Significant progress has been made in understanding the life cycle of convectively generated "anvil" cirrus and the response of their macrophysical properties to large-scale controls. On the other hand, much work remains to be done to fully understand how small-scale anvil processes and the climatological anvil radiative effect will respond to global warming. Thin, in situ-formed cirrus are now known to be closely tied to the thermal structure and humidity of the tropical tropopause layer, but microphysical uncertainties prevent a full understanding of this link, as well as the precise amount of water vapor entering the stratosphere. Model representation of ice-nucleating particles, water vapor supersaturation, and ice depositional growth continue to pose great challenges to cirrus modeling. We believe that major advances in the understanding of tropical cirrus can be made through a combination of cross-tool synthesis and cross-scale studies conducted by cross-disciplinary research teams.

## 1  Introduction

The tropical upper troposphere is the cloudiest place on Earth due to widespread cirrus cloud coverage (Fig. 1). These cirrus arise from two main sources: deep convection and in situ ice formation outside of convective clouds. Anvil cirrus form when

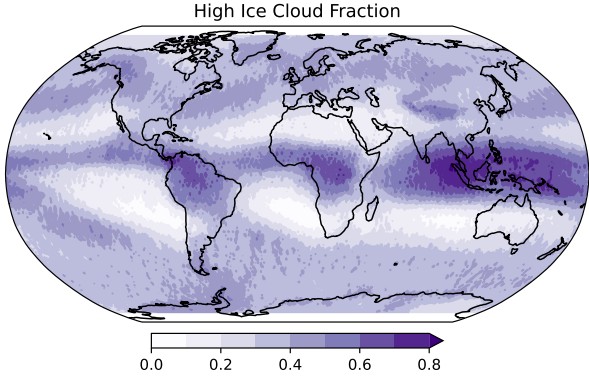

**Figure 1.** Cirrus cloud fraction observed by the CALIOP lidar from 2007-2015. Cirrus clouds are here defined as those with cloud top pressure below 400 hPa, including deep convective clouds. Data were obtained from the CALIPSO Level 3 GEWEX Cloud product, v1-00 NASA/LARC/SD/ASDC (2019)

deep convection deposits moist and ice-laden air in the upper troposphere. Often outliving the deep convective systems that generate them, anvils cover up to one-third of convectively active regions such as the Indo-Pacific Warm Pool and the Intertropical Convergence Zone (ITCZ) (Yuan and Houze, 2010). Above the typical level of deep convective outflow, in the tropical

tropopause layer (TTL; $\simeq$14–19 km), cirrus formed in situ are ubiquitous (Fig. 2). Here, the large-scale ascent characteristic of the TTL drives adiabatic cooling and high relative humidity, creating a favorable environment for ice formation (Fueglistaler et al., 2009; Randel and Jensen, 2013).

Tropical cirrus clouds encompass both anvil and TTL cirrus and are of critical importance to many aspects of Earth's climate system. The climatological significance of anvil cirrus begins with their influence on radiative fluxes, which leads not only to

large, local changes in the top-of-atmosphere energy budget, but also to substantial diabatic heating rates within the atmosphere itself (Robert A Houze, 1982; Ackerman et al., 1988; Ramanathan et al., 1989). These cloud radiative effects (CRE) play an important role in many aspects of global circulation and climate (Fig. 2). By decreasing the net radiative cooling rate of the atmosphere below, anvil cirrus reduce the demand for latent heat in the troposphere, which slows the hydrologic cycle (Su et al., 2017; Albern et al., 2018). Anvils are thought to contribute to the narrowing of the ITCZ, broadening of regions of

subtropical descent, and strengthening of the Hadley cell (Harrop and Hartmann, 2016; Popp and Silvers, 2017; Albern et al., 2018). Their radiative impacts have been identified as a critical component of the Madden-Julian Oscillation (Raymond, 2001) and of convective self-aggregation in high-resolution numerical models (Wing and Emanuel, 2014; Holloway and Woolnough, 2016).

While thin in situ cirrus in the TTL also modify local radiative fluxes (Haladay and Stephens, 2009), their global importance

arises primarily from their effects on the transport of water vapor into the stratosphere. Ascent across the tropical tropopause is the primary source of tropospheric air to the stratosphere (Holton et al., 1995). As air ascends through the TTL (Fig. 2), its water content is depleted as ice crystals nucleate, grow via vapor deposition, and sediment downward, removing water to lower levels

(Jensen et al., 1996). Following this freeze-drying, tropical stratospheric air is transported poleward by the Brewer-Dobson circulation. Cirrus microphysical processes thus regulate global stratospheric water vapor content, with important implications for stratospheric chemistry and Earth's radiation budget (Solomon et al., 2010).

Because of these global impacts, accurate understanding and characterization of tropical cirrus is critical to model climate and project its changes. While we have expanded satellite and high-altitude aircraft datasets and improved ice microphysics parameterizations over the last two decades, our understanding of tropical cirrus remains incomplete. Indeed, changes in anvil cirrus areal coverage with warming were the least certain climate feedback in a recent assessment of Earth's climate sensitivity (Sherwood et al., 2020). Much of this uncertainty stems from the fact that cirrus are notoriously difficult to model (Sec. 5) and to observe for both in situ- and satellite-based instruments. From ice microphysics to planetary scale circulations, cirrus are sensitive to processes occurring across a wide range of spatiotemporal scales, as discussed below (Fig. 3).

This opinion piece has several goals. One is to synthesize recent progress in cirrus studies, both its methodologies and its important findings. We also articulate the questions that present the most urgent challenges to our understanding of tropical cirrus and their role in global climate and climate change. Finally, we identify promising ways forward.

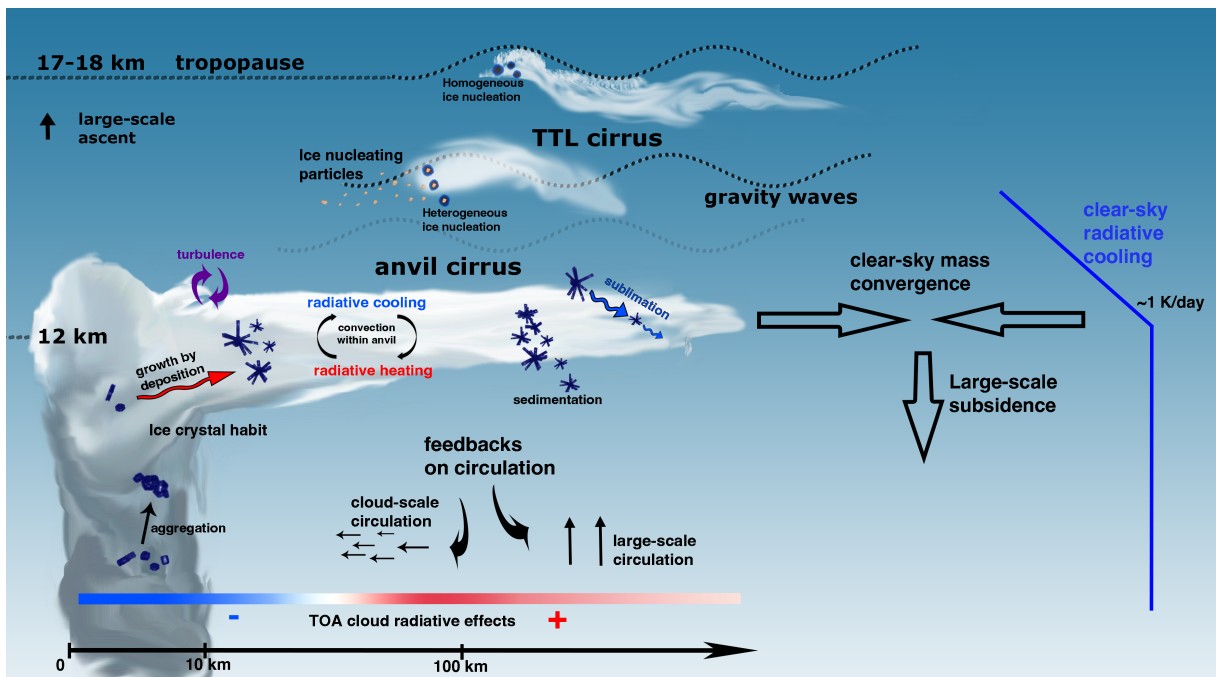

**Figure 2.** Illustration of key processes controlling tropical anvil cirrus and TTL cirrus. These include microphysical processes such as ice nucleation, depositional growth and sublimation, aggregation, and sedimentation, interaction of ice crystals with radiative fluxes, and large-scale mass convergence and subsidence driven primarily by clear-sky radiative cooling, large-scale ascent in the TTL, and gravity waves. The interaction of small-scale with large scale processes in anvils can drive circulations and generate turbulence. Processes with a significant influence on the diabatic temperature heating budget are colored in red (warming) or blue (cooling).

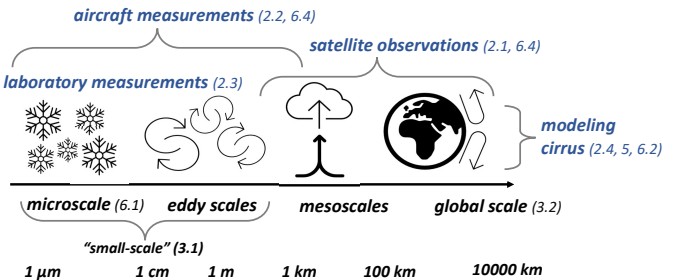

**Figure 3.** Range of scales defining cirrus properties and evolution and methods to probe these different scales. We include references to article subsections with discussions of these scales and methods.

## 2 How cirrus are studied

### 2.1 Observing cirrus from space

Satellite measurements provide a unique global perspective on tropical cirrus clouds, adding to the information from ground-based and aircraft measurements. We divide these remote sensing measurements into polar-orbiting active, polar-orbiting passive, geostationary, tracking, and synergistic datasets.

Several spaceborne active sensors provide advanced understanding of tropical cirrus properties, evolution, and radiative effects. Launched in 2006, the CloudSat Cloud Profiling Radar (Stephens et al., 2008) and the Cloud-Aerosol Lidar with Orthogonal Polarization (CALIOP) (Winker et al., 2010) represented a substantial increase in the observational capacity for high clouds and their properties (Stephens et al., 2018). This combination of active instruments provides global observations of cloud fraction and condensate mixing ratio profiles for both optically thick and thin clouds. Using the merged CloudSat radar reflectivity and CALIOP lidar attenuated backscatter signals, the CloudSat level 2C Ice Cloud Property Product (2C-ICE, Deng et al., 2010) and the radar/lidar product (DARDAR, Delanoë and Hogan, 2008, 2010) retrieve ice water content, effective ice crystal size, and extinction coefficient. Sourdeval et al. (2018) extended the DARDAR algorithm to globally compute ice crystal number concentration, a quantity that can vary by orders of magnitude and thus better constrain ice nucleation pathways (Gryspeerdt et al., 2018), aerosol-cirrus interactions (Gryspeerdt et al., 2018; Mitchell et al., 2018; Zhu et al., 2022) and anvil-radiative feedbacks (Sokol and Hartmann, 2020; Wall et al., 2020). CloudSat-CALIPSO-derived cirrus properties were also used as inputs to estimate cloud radiative heating rates within the atmosphere (L'Ecuyer et al., 2008; Henderson et al., 2013; Matus and L'Ecuyer, 2017; Hang et al., 2019) and cloud radiative effects (CREs) at the top and bottom of the atmosphere (Hong et al., 2016; Matus and L'Ecuyer, 2017; L'Ecuyer et al., 2019). The high sensitivity of CALIOP lidar retrievals to small particles (Winker et al., 2010; Avery et al., 2012) has greatly enhanced knowledge of the occurrence and radiative effects of TTL cirrus (Sassen et al., 2009; Massie et al., 2010; Haladay and Stephens, 2009; Yang et al., 2010; Fu et al., 2018), their interannual variability (Virts et al., 2010; Tseng and Fu, 2017), and their relationship to gravity waves (Chang and L'Ecuyer, 2020).

Despite supplying vertical profiles of cloud properties, active instruments suffer from poor statistical sampling: they only provide a curtain of measurements along the satellite path. Passive instruments overcome the sampling problem with their wide swathes, but often sacrifice spatial resolution as a result. Hence, passive instruments, such as the Atmospheric Infrared Sounder (AIRS, Chahine et al., 2006) or the Infrared Atmospheric Sounding Interferometer (IASI, Hilton et al., 2012), give statistically robust estimates of cloud fraction and other macrophysical properties over the full globe (Stubenrauch et al., 2017; Leonarski et al., 2020). These macrophysical observations have previously been combined with meteorological data to study the response of tropical cirrus to changes in surface temperatures (Zelinka and Hartmann, 2011; Stubenrauch et al., 2017; Protopapadaki et al., 2017) and to constrain global-climate model (GCM) ice microphysics (Stubenrauch et al., 2019). Recently, a combination of multiangle polarization and shortwave infrared measurements of Polarization and Directionality of the Earth's Reflectances (POLDER) and Moderate Resolution Imaging Spectroradiometer (MODIS) data provided information on ice crystal properties at the tops of thick ice clouds (van Diedenhoven et al., 2016; van Diedenhoven et al., 2020).

Geostationary satellites make up another class of remote sensing data, particularly useful in estimating high cloud evolution due to their continuous, high-frequency sampling. Cirrus clouds have been observed for decades as part of the International Satellite Cloud Climatology Project (ISCCP) (Rossow and Schiffer, 1999), which combines visible and infrared measurements from multiple geostationary and polar-orbiting satellites with sub-daily resolution. These measurements provide information on cloud amounts, cloud particle effective size, cloud optical depth, and cloud top temperatures (Parol et al., 1991; Rossow and Schiffer, 1999; Stubenrauch et al., 2013). ISCCP data have been used to validate high clouds in model simulations (Pincus et al., 2012; Tselioudis et al., 2021) and to study the lifecycle and diurnal cycle of tropical cirrus (Luo and Rossow, 2004; Wylie and Woolf, 2002; Yin and Porporato, 2017). The multidecadal duration of ISCCP data provided the first evidence for altitude shifts of high clouds with warming (Norris et al., 2016).

Geostationary observations are also used to track mesoscale convective systems (MCSs), organized storms whose outflow is the source of tropical anvil cirrus (e.g., Fiolleau and Roca, 2013; Roca et al., 2017). From visible and infrared channels, a transition can be inferred from isolated convective cores to optically thick anvil outflow to optically thin residual outflow in the growing, mature, and dissipating stages of an MCS lifecycle (Jones et al., 2023). This inference is generally based upon brightness temperature thresholds or changes in cloud-top radiative cooling rates. Combining geostationary infrared imagery with environmental and radiative properties from low-earth orbiting satellites or from satellite-based precipitation measurements has also provided insight into the MCS and anvil lifecycles (e.g., Feng et al., 2011; Fiolleau and Roca, 2013). Importantly, Feng et al. (2021) have developed a global MCS tracking database at hourly and 10-km resolution using a fusion of geostationary satellite data and Global Precipitation Measurement mission rainfall observations. An important challenge for the community will be standardizing various tracking databases.

Finally, remote sensing datasets that combine multiple satellite observations and reanalysis fields are of particular utility for careful studies of tropical cirrus properties and their radiative effects (e.g., Wall et al., 2018). CALIPSO-CloudSat-CERES-MODIS Merged Product (CCCM, Kato et al., 2011) represents such a dataset, frequently used for model evaluation purposes (Xie et al., 2018; Caldwell et al., 2021).

## 2.2 In situ measurements

Relative to satellite measurements, in situ data have better spatio-temporal sampling and more precise measurements of cirrus microphysical evolution and vertical distribution. In situ measurements are the only way to obtain detailed information about the size distributions and habits of cirrus ice crystals and ice-nucleating particle (INP) size and chemistry. The in situ measurements not only provide information about cirrus microscale properties and physical processes, they allow evaluation and improvement of the satellite remote-sensing retrieval algorithms. On the other hand, in situ aircraft data are limited by the inability to sample at close proximity to convective cores and a far smaller sample size than satellites.

Several field campaigns have probed tropical cirrus over the past few decades. In the early 2000s, the Cirrus Regional Study of Tropical Anvils and Cirrus Layers Florida Area Cirrus Experiment (CRYSTAL-FACE; Jensen et al., 2004) took place out of Florida, providing ice crystal, radiative, and meteorological measurements. These measurements provided the basis for early aerosol-dependent ice nucleation parameterizations (Phillips et al., 2008). In 2006, the Tropical Warm Pool-International Cloud Experiment (TWP-ICE) again focused on providing observational constraints for ice cloud parameterizations with flights out of Darwin, Australia (May et al., 2008). During the same year, in situ measurements were made in anvil cirrus generated by MCSs over west Africa during the Stratospheric-Climate Links with Emphasis on the UTLS - African Monsoon Multidisciplinary Analysis campaign (SCOUT-AMMA; Frey et al., 2011). The 2006 Costa Rica Aura Validation Experiment (CR-AVE) and the 2007 Tropical Composition, Cloud and Climate Coupling Experiment (TC4) extensively sampled anvil cirrus over the eastern tropical Pacific (Jensen et al., 2009). Four years later, the Ice In Clouds Experiment-Tropical (ICE-T), operating out of the U.S. Virgin Islands, studied ice nucleation pathways in tropical cirrus (Heymsfield and Willis, 2014). In 2014 and 2015, the High Altitude Ice Crystals-High Ice Water Content (HAIC-HIWC) campaign sampled both oceanic and continental deep convection and cirrus outflow near Darwin, Australia, Cayenne, French Guiana, and Fort Lauderdale, USA (Hu et al., 2021a; Strapp et al., 2020). The Airborne Tropical TRopopause EXperiment (ATTREX; Jensen et al., 2017) and POSIDON (Pacific Oxidants, Sulfur, Ice, Dehydration, and cONvection experimen; Jensen et al., 2018a) campaigns provided extensive measurements of TTL cirrus, as well as anvil cirrus generated by typhoons. Both the Stratospheric and Upper Tropospheric Processes for Better Climate Predictions (StratoClim) and Asian Summer Monsoon Chemical and Climate Impact Project (ACCLIP) campaigns yielded new sources of upper-tropospheric ice crystal size distributions, water vapor isotopic composition (a proxy for ice crystal origin), and specific humidity during the Asian monsoon time frame. Airborne campaign measurements have also been used to develop the vast majority of parameterizations used in satellite retrieval algorithms.

A good summary of these tropical cirrus-focused field campaigns exists in the "Cirrus Guide" of Krämer et al. (2016) and Krämer et al. (2020). These guides aggregate the in situ measurement statistics to construct ice water path-temperature and ice crystal number-temperature spaces from 24 field campaigns conducted between 1999 and 2017 with a total sampling time in cirrus of 168 hours, two-thirds of which took place in the tropics. These climatologies of ice cloud properties versus temperature find that tropical cirrus tend to be optically thicker more often than their midlatitude counterparts. A large amount of in-situ observations of tropical anvil clouds is also available in the HAIC-HIWC campaign dataset (Strapp et al., 2020).

Weather balloons and quasi-Lagrangian super-pressure balloons are additional platforms to collect in situ data on cirrus and the upper troposphere (Cirisan et al., 2014; Khaykin et al., 2016; Wolf et al., 2018) and upper tropospheric waves (Podglajen et al., 2016; Corcos et al., 2021). Balloons circulating the tropics at an altitude of 18.5 and 20.5 km were deployed as part of the Stratéole 2 field campaign (Haase et al., 2018; Kalnajs et al., 2021). These balloons carry a number of lightweight instruments,
such as a reel-down instrument for in situ measurements of TTL cirrus down to 2 km below the balloon flight level, providing information on the interaction of equatorial waves with TTL cirrus (Bramberger et al., 2022). Some of the balloons carry a microlidar system that can detect even the thinnest TTL cirrus (Ravetta et al., 2020). Dynamical information inferred from balloons has already been used in several studies of TTL cirrus (Jensen et al., 2016; Corcos et al., 2023), and offers further insight into the tropical cirrus life cycle 6.3.

**2.3   Laboratory measurements**

Laboratory studies are critical for advancing fundamental understanding of cloud microphysical processes (Morrison et al., 2020). Insights from theory, models, or field observations can be validated against well-designed laboratory measurements. To develop process-level understanding and to identify key parameters of the processes under study, experiments under well-controlled conditions are critical.
Examples of laboratory work related to cirrus include study of thresholds and rate constants for homogeneous ice nucleation from solution droplets (Koop et al., 2000; Schneider et al., 2021), as well as the environmental conditions under which various INPs contribute to ice formation (see Kanji et al. (2017) for a recent review). As opposed to the macrophysical cloud scales (order of km) retrieved by satellites and simulated by models, cloud microphysical processes occur at the molecular scale. Recent ice nucleation studies highlight the importance of the nanometer scale with direct observation of initial ice germ
formation along inhomogeneities in the INP surface (Kiselev et al., 2017). Nanometer-sized pores were also found to be an important factor driving ice nucleation in cirrus clouds (Marcolli, 2014; David et al., 2019).

Laboratory studies have also been critical in developing a fundamental understanding of ice crystal growth (Libbrecht, 2005). Ice crystal growth from water vapor is known to be sensitive to crystal surface properties (Magee et al., 2014) and the ambient temperature and supersaturation. Many laboratory studies have tried to quantify the water vapor deposition coefficient (Skrotzki
et al., 2013; Lamb et al., 2023), which influences cirrus cloud microphysical and radiative properties. Moreover, changes in the preferential partitioning of water isotopes by non-equilibrium kinematic effects during ice deposition gives insight into ice crystal surface kinetics and deposition coefficients (Nelson, 2011; Lamb et al., 2017).

While studies of INPs and depositional growth can be performed with benchtop setups, larger cloud-chamber facilities are better able to replicate real-world conditions for cirrus, in part by minimizing wall effects and strong thermodynamic gradients
and in part by providing more robust statistics from larger sample sizes. Because of significant installation and maintenance costs, relatively few large cloud chambers exist in the world. The Aerosol Interaction and Dynamics in the Atmosphere (AIDA) chamber at the Karlsruhe Institute of Technology and the Pi Cloud Chamber at Michigan Tech are two such facilities that have increased our understanding of 1) the deposition coefficient of water vapor molecules, 2) the effect of organic coatings on ice-

nucleating efficiencies and, 3) secondary ice nucleation in the wake of sedimenting crystals, among other processes (Skrotzki et al., 2013; Kanji et al., 2019; Prabhakaran et al., 2020).

However, even for larger-scale chamber experiments, it often remains unclear to what extent results can be directly compared to atmospheric conditions. For example, while laboratory studies use ice crystals produced in a controlled pure water vapor environment, atmospheric aging processes can alter both ice crystal nucleation and growth mechanisms. These limitations suggest that laboratory experiments should be combined with satellite and in situ observations to improve our knowledge of tropical cirrus.

## 2.4 Modeling cirrus

Numerical models represent another class of tools used to study tropical cirrus. The wide range of spatial and temporal scales involved in tropical cirrus (Figs. 2 and 3) means that no single modeling tool can describe all processes and interactions. Complicating this situation further is the need to account for the small-scale dynamical forcings that drive ice formation and for detailed aerosol information, e.g., size distributions and chemical composition of INPs (Sec. 5.1), to enable the calculation of aerosol-cirrus interactions.

Parameterizations are simplified representations of physical processes that focus on a limited number of interactions and require validation with detailed numerical models. Microphysical parameterizations with varying degrees of complexity have been developed for ice nucleation, depositional growth, and other subgrid-scale processes (e.g., Hubbard, 1977; Barahona and Nenes, 2008, 2009; Connolly et al., 2012; Phillips et al., 2008; Kärcher and Lohmann, 2002, 2003). Radiation parameterizations are another example, wherein the representation of unresolved variability in cirrus properties on resolved mean fluxes and heating rates remains challenging. An important but often overlooked factor is the need for microphysical and radiation parameterizations to have similar degrees of complexity and consistent formulations, i.e., the same definitions of commonly used variables and same degrees of approximation (Baran et al., 2014, 2016; Sullivan and Voigt, 2021).

These parameterizations are at the heart of GCMs that run at the coarsest spatial and temporal scales. With increasing computational power, grid spacings can be greatly refined, and global storm-resolving models (GSRMs) now operate at kilometer-scale resolution, reducing the number of subgrid processes that must be parameterized. In particular, GSRMs resolve deep convection explicitly, removing numerous longstanding biases caused by convective parameterization (Stevens et al., 2020; Judt et al., 2021; Feng et al., 2023). They bridge the gap between GCMs and classic cloud-resolving models (CRMs) and allow improved simulation of cirrus evolution in the presence of realistic large-scale circulations.

High-resolution CRMs that are capable of simulating anvil life cycles are used for validation and advanced process studies (Saleeby and van den Heever, 2013; Gasparini et al., 2019; Prein et al., 2021). Regional-scale CRMs capture more scale-dependent interactions and some large-scale controls, but still rely on simplified microphysical schemes representing bulk cloud properties only (Powell et al., 2012; Weverberg et al., 2013). Along with full-field simulations that represent an Eulerian perspective of ice clouds, trajectory models can also be used for a Lagrangian perspective (Sec. 6.3). They may discretize ice crystal growth over a fixed-size grid or track individual ice particles.

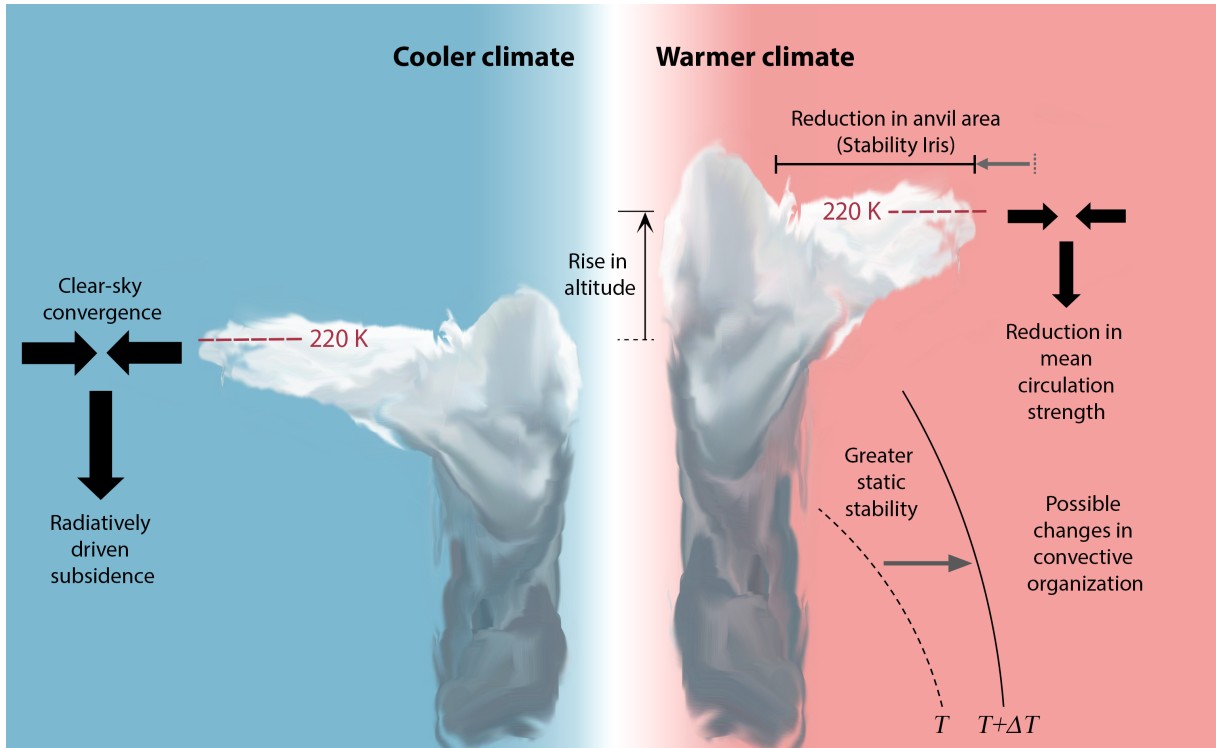

**Figure 4.** Schematic of large-scale controls on anvil cirrus properties and their hypothesized responses to global warming.

## 3 Anvil cirrus

Anvil clouds are unique among Earth's clouds because they can exert both positive and negative CRE over their life cycle. Anvil clouds form at the top of the convectively mixed troposphere, near the 220-K ($\sim -55°$ C) isotherm, where the water vapor radiative cooling rapidly declines (Fig. 2 and Sec. 3.2.1). They can be optically thick and precipitating at the time of their formation, but over their life cycle various dynamical, radiative, and microphysical processes cause them to thin and spread out over a large area (Lilly, 1988; Ackerman et al., 1988; Hartmann et al., 2018; Schmidt and Garrett, 2013; Gasparini et al., 2019, 2021; Wall et al., 2020; Garrett et al., 2005). Cloud ice is slow to sublimate at the frigid temperatures of the tropical upper troposphere (Seeley et al., 2019), and diabatically driven circulations can sustain cloud ice against sedimentation (Hartmann et al., 2018; Gasparini et al., 2022). These processes, among others, allow anvil cirrus to cover a much greater area and persist for much longer periods of time than deep convective towers themselves (Luo and Rossow, 2004).

Because of their convective origins, anvil cirrus may be affected by any factor that affects the distribution, intensity, and character of deep convection. As a result, anvils are linked to processes across a wide range of spatial and temporal scales. Large-scale tropical oscillations such as ENSO, the Madden-Julian Oscillation, and the Quasi-Biennial Oscillation may impact the distribution, properties, and overall abundance of anvil cirrus (Riley et al., 2011; Sullivan et al., 2019; Sweeney et al., 2023). At the same time, the evolution of individual anvil cloud systems is shaped by mesoscale and cloud-scale circulations. These

circulations are driven by radiative and latent heating, both of which depend critically on microphysical processes (Gasparini et al., 2019). All of these scales must be considered to fully understand the role of anvil cirrus in global climate. This section discusses progress and challenges in this important area of work. We first address small-scale processes affecting anvil cirrus evolution before turning to large-scale controls and anvil radiative feedbacks.

## 3.1  Small-scale processes in anvils and their implications

### 3.1.1  Which microphysical processes occur in anvil cirrus life cycle?

Deep convective clouds form at the top of the boundary layer, and the first ice crystals begin to form by heterogeneous nucleation between 6 and 9 km altitude, roughly between $-10$ and $-30$ °C. Some liquid droplets, however, may persist up to the homogeneous freezing temperature of water ($-38$ °C) at about 10 km (Rosenfeld and Woodley, 2000). In recent years, much effort has been devoted to identifying which aerosols serve as INPs during convective ascent (Sec. 5.1), and several other reviews are devoted to summarizing this work (Kanji et al., 2017; Murray et al., 2021). As they ascend, ice crystals grow by deposition, a process whose rate is affected by crystal non-sphericity and remains challenging to represent (Sec. 5.3). A fraction of the large ice crystals formed via heterogeneous nucleation is likely to break up and produce a large number of smaller ice crystals—a process known as secondary ice formation (Ladino et al., 2017; Qu et al., 2022). Remote sensing observations and exploratory simulations generally indicate that secondary ice production can enhance ice crystal numbers 1000-fold but with infrequent occurrence ($<10\%$) (Luke et al., 2021). The relative importance of primary nucleation versus secondary production as ice crystal sources in anvil clouds is still subject to large uncertainties (E. Hawker et al., 2021; Sullivan et al., 2017, 2018).

When convecting air parcels full of ice reach the level of neutral buoyancy, the air is injected into a stably stratified environment, which leads to flattening and spreading of this air and formation of a thick anvil cirrus cloud (Lilly, 1988). Within a few hours of detrainment, the largest ice crystals quickly precipitate out of the cloud and sublimate in the subsaturated layer below, reducing the ice content and optical depth of the anvil (Jensen et al., 2018b; Gasparini et al., 2021). The cloud loses additional mass by forming larger, rapidly falling snowflakes in a process called aggregation, which is relevant at temperatures warmer than $-40°$ C. Falling snowflakes can collect ice crystals through accretion, further decreasing the anvil ice water path (Heikenfeld et al., 2019).

As the anvil reaches its mature stage, cloud-radiation interactions as well as latent heating and cooling begin to play an important role in modulating its morphology and climatic effects (Sec. 3.1.2). These effects frequently manifest at the cloud- and mesoscale (Dinh et al., 2023), but can also impact large-scale circulation in the tropics and extratropics (Schumacher et al., 2004; Voigt and Shaw, 2016; Voigt et al., 2019) (Sec. 3.1.3). Finally, sedimentation and sublimation dominate the decaying stage of the cloud. The decay timescale is therefore strongly dependent on the temperature (colder temperatures imply slower sublimation, Seeley et al., 2019), environmental relative humidity and size of ice crystals (Jensen et al., 2018b), as well as the mesoscale and synoptic-scale dynamical support.

### 3.1.2 How do microphysical processes determine diabatic heating?

The processes detailed in the previous section influence atmospheric temperature and pressure gradients through their associated latent and radiative heating. Micro- and mesoscale cloud radiative effects play an important role in the anvil cirrus lifecycle. Latent heating dominates the contribution to cloud-scale circulation over radiative heating, especially early in the cloud life cycle and in the lower part of the cirrus outflow (Gasparini et al., 2019).

As noted before, tropical cirrus can generate positive or negative radiative effects at the top of atmosphere or surface based on their optical thickness and age. Microphysical processes and parameters play an important role in this modulation (Zhang et al., 1999). Modeling studies show that ice microphysics schemes can alter the top-of-the-atmosphere outgoing longwave radiation by as much as 30 W m$^{-2}$ while the consideration of different ice habits can alter ice crystal number concentrations and resultant radiative effects drastically (Sullivan and Voigt, 2021; Lamraoui et al., 2023). Along with the small-scale physics and shape of ice crystals, their complexity—a term encompassing factors such as surface roughness and hollowness—can generate additional shortwave (SW) cooling on the order of 1 W m$^{-2}$ (Järvinen et al., 2018). Cirrus optical properties can also alter top-of-the-atmosphere and surface fluxes by as much as 10s of W m$^{-2}$ (Zhao et al., 2018; Yi, 2022).

Microphysical processes also influence—and are influenced by—the diurnal cycle of these fluxes. SW absorption by crystals produces a mesoscale circulation that buoys the cirrus and prolongs its lifetime (Sec. 3.1.3). Polarimetric difference measurements of the Global Precipitation Measurement Microwave Imager indicate that ice crystal size and shape have strong diurnal variations over land and are related to the cycles in macroscopic variables, such as vertical or horizontal extent (Gong et al., 2017). Flux divergences, or atmospheric cloud-radiative heating rates, have been relatively less studied than the top-of-the-atmosphere or surface fluxes. But cloud radiative heating, especially in the upper troposphere, has a more direct influence on circulation, and is modulated strongly by ice microphysical factors such as the aerosol dependence of the nucleation scheme or the consistency between the microphysical and optical parameterizations (Sullivan and Voigt, 2021; Sullivan et al., 2022). Cloud radiative heating climatologies and ensembles, both observational and model-based, indicate the largest differences in the tropical upper troposphere (Voigt et al., 2019; Cesana et al., 2019), and the associated ice microphysical controls warrant further study.

### 3.1.3 How are anvil cirrus circulations affected by diabatic heating?

The interaction of ice crystals with radiation in anvil cirrus leads to changes in upper tropospheric temperature gradients, which drive large-scale and mesoscale circulations. For brevity, we omit the cirrus impacts on large-scale circulations in this Opinion piece (interested readers can find more in Voigt et al., 2019, 2021). Typically, mesoscale circulations consist of an updraft within the radiatively heated cloud with horizontal outflow near cloud top and compensating sinking motion in nearby clear-sky areas. The circulation is closed by inflow near cloud base, where the maximum radiative heating is located (Ackerman et al., 1988; Durran et al., 2009; Dinh et al., 2010; Gasparini et al., 2019; Dinh et al., 2023). Such circulations are boosted by SW heating during daytime; in-cloud updrafts around the middle of the day, when the insolation is high, can be strong enough to counteract the sedimentation of small ice crystals, prolonging cloud lifetime (Deng and Mace, 2008; Wall et al., 2020;

Gasparini et al., 2022). Circulation patterns can be more complex in anvil cirrus of optical depths larger than approximately 3 with strong cloud top cooling due to emission of longwave radiation, which induces an additional circulation cell. This consists of a sinking inflow of drier environmental air that erodes the cloud top and decreases the lifetime of the cloud (Gasparini et al., 2022).

Prolonged exposure to intense cloud heating can cause isentropic surfaces to warp, leading to development of unstable, turbulent, convective layers in high clouds (Dobbie and Jonas, 2001; Dinh et al., 2010; Ferlay et al., 2014; Schmidt and Garrett, 2013). Radiatively driven turbulence within anvils can occasionally trigger new ice crystal nucleation, prolonging the lifetime of the cloud and influencing the optical depth distribution of anvils (Hartmann et al., 2018; Sokol and Hartmann, 2020).

It is currently not known whether and how the interactions between microphysics, radiation, latent heating, and the resulting circulations may change in a warmer world and whether this contributes substantially to the uncertainty in anvil cirrus feedbacks (as outlined in Section 3.2).

## 3.2 Large scale anvil cloud climate feedbacks

### 3.2.1 What controls the altitude of anvil cirrus?

It can be argued on the basis of radiative-convective equilibrium physics that the tops of convective clouds should remain at about the same temperature as the climate changes (Hartmann and Larson, 2002). To obtain this result it must be assumed that water vapor is the dominant agent for cooling the atmosphere, in which case the ability of the atmosphere to cool and thereby balance convective heating declines rapidly at a temperature of about 220 K ($\sim -55$ °C). This marks the top of the actively convecting layer of the atmosphere (Fig. 2) and is where the strongest net detrainment of cloud ice occurs (Kuang and Hartmann, 2007; Harrop and Hartmann, 2012; Hartmann et al., 2019; Jeevanjee and Fueglistaler, 2020). If the highest cloud tops do remain at a constant temperature as the climate changes, this Fixed-Anvil Temperature (FAT) produces a climate with a more positive (global warming intensifying) radiative feedback than if the clouds remained at a fixed pressure, because the emission from an optically thick cloud top remains unchanged as the surface temperature warms, decoupling the upward longwave cloud emission from the surface temperature (Zelinka and Hartmann, 2010). The assignment of a positive feedback to this FAT mechanism is complicated by the fact that the emission temperature of the clear atmosphere in which the clouds are embedded also tends to remain constant as the climate changes for the same physical reasons on which the FAT hypothesis is based (Koll and Cronin, 2018; Stevens and Kluft, 2023).

The FAT hypothesis of fixed cloud top temperatures assumes the dominance of water vapor for atmospheric radiative cooling and equilibration of clouds and climate, as over the warm oceans. If these assumptions are violated as the climate warms, the average cloud top temperature can vary. For example, if the 220K isotherm extends upward into the region of the atmosphere where absorption of solar radiation by ozone becomes important, then the cloud tops will warm as the surface temperature increases (Thuburn and Craig, 2002; Kuang and Hartmann, 2007; Harrop and Hartmann, 2012; Seidel and Yang, 2022). We know that individual convective cells do penetrate above the 220K level and occasionally overshoot beyond the tropopause. Over land with a strong diurnal cycle, the convection may be out of equilibrium with the global tropics and cloud tops may

penetrate beyond the temperature threshold predicted by the FAT hypothesis. Average cloud top temperatures are different over the western and eastern Pacific ocean, although the relationship between cloud abundance and radiatively-driven large-scale divergence seems to hold (Kubar et al., 2007). The radiative cooling rate at a fixed temperature can vary depending on the lapse rate (Hartmann et al., 2022), so that if lapse rates change with the climate, then we might expect cloud top temperature to also change. Tropical cloud properties over land areas are different from those over ocean areas, with updraft velocities over land exceeding those over the oceans (Liu et al., 2007). As the climate changes, the distribution of sea-surface temperature and the contrast between land- and sea-surface temperature may shift, in turn altering the average cloud top temperature and Earth's energy balance. The response of anvil cirrus temperature and amount to changes in the geographical distribution of surface temperature should be a key topic for ongoing research.

### 3.2.2 What controls the amount of anvil cirrus on large scales?

Changes in anvil cirrus cloud amount (ACA) and their associated climate implications have been a subject of much interest since the publication of the Iris hypothesis over 20 years ago (Lindzen et al., 2001). Based on 20 months of satellite observations, this original study found ACA to be inversely related to sea surface temperature (SST). While controversial at the time of its publication (Hartmann and Michelsen, 2002; Fu et al., 2002), the Iris hypothesis raised the important questions of whether changes in ACA could act as a significant feedback on global warming. Today, this question remains relevant and without consensus.

The study by Lindzen et al. was the first of many seeking to understand the ACA-SST relationship by leveraging interannual climate variability over the satellite record (Su et al., 2008; Zelinka and Hartmann, 2011; Behrangi et al., 2012; Igel et al., 2014; Zhou et al., 2014; Choi et al., 2017; Liu et al., 2017; Kubar and Jiang, 2019; Saint-Lu et al., 2020, 2022; Höjgård-Olsen et al., 2022). These studies have produced varied results, which may be expected due to differences in study region and period, satellite instrumentation, how ACA is quantified, and how, if at all, anvil cirrus are distinguished from TTL cirrus and deep convective cores. Nevertheless, on the whole, these observational assessments suggest an inverse relationship between tropical SST and ACA, although the strength of this relationship likely varies from region to region and as a function of anvil cirrus thickness.

Most CRMs and GCMs also suggest that ACA decreases with warming (Zelinka and Hartmann, 2010; Wing et al., 2020; Stauffer and Wing, 2022), albeit with less consistency than observations of interannual variability. Increases in ACA with warming were found in one-third of the models participating in the Radiative-Convective Equilibrium Model Intercomparison Project (RCEMIP; Wing et al., 2018, 2020) and have long been a salient feature of NICAM, a GSRM (Tsushima et al., 2014). These inconclusive results, along with the known sensitivity of simulated anvil cirrus to factors such as horizontal resolution (Jeevanjee and Zhou, 2022) and ice crystal habit assumptions (Lamraoui et al., 2023), show just how difficult anvil cirrus are to model (see Section 5).

In addition to these assessments, significant progress has been made in understanding the physical mechanisms underlying the ACA response to warming. The original Iris hypothesis speculated that the decline in ACA with increasing SST was caused by the enhancement of precipitation efficiency (PE) within convective updrafts (Lindzen et al., 2001). While this mechanism

has been demonstrated in models (Mauritsen and Stevens, 2015; Jeevanjee and Zhou, 2022; Li et al., 2019), it is difficult to assess in the real world because PE, as is it traditionally defined, is a difficult quantity to observe on a large scale. Studies using modified definitions of PE suited for observational analyses have yielded conflicting results (Ito and Masunaga, 2022; Choi et al., 2017). Moreover, whether or not PE actually increases in response to warming remains uncertain.

An alternative explanation relates ACA to the overall strength of the convective overturning circulation in the Tropics, which has long been expected to weaken with warming (Betts and Ridgway, 1989; Knutson and Manabe, 1995; Held and Soden, 2006). Several explanations have been provided for this weakening (see Jenney et al., 2020 for a thorough discussion), but one—the Stability Iris mechanism outlined in Bony et al. (2016)—has come to dominate the discussion of ACA (Fig. 4). In brief, the Stability Iris hypothesis states that surface warming causes the tropical troposphere to become more stable, which leads to a reduction in the radiatively driven, clear-sky convergence at the anvil cloud level (Zelinka and Hartmann, 2010). Because clear-sky convergence must be balanced by the detrainment of mass from deep convection, a reduction in clear-sky convergence leads to reduced detrainment and reduced ACA. Observations (Zelinka and Hartmann, 2011; Saint-Lu et al., 2020, 2022) and models (Stauffer and Wing, 2022) are generally consistent with the Stability Iris hypothesis, although a small number of the RCEMIP models produce simultaneous reductions in clear-sky convergence and increases in ACA in apparent violation of Stability Iris physics (Stauffer and Wing, 2022). While the Stability Iris hypothesis represents significant progress and may very well describe the first-order impact of warming on ACA, other work suggests that there is more to the story (Beydoun et al., 2021; Jeevanjee, 2022). This is hardly surprising considering that anvil cirrus dynamics are affected by processes occurring on much smaller scales than those addressed by Stability Iris physics (section 3.1.1).

Another factor that is known to affect ACA is the large-scale aggregation of convection. Observations (Yuan and Houze, 2010; Tobin et al., 2012, 2013; Stein et al., 2017) and models (Bony et al., 2016; Wing and Cronin, 2016; Wing et al., 2020) both show that higher degrees of convective aggregation are associated with reduced ACA. This link appears to be robust, suggesting that changes in aggregation could modulate the ACA response to warming. Unfortunately, convection-resolving models do not provide a consensus on how warming impacts aggregation (Wing et al., 2020); even if they did, model-simulated aggregation is not exactly the same as the forms of convective organization observed in the Tropics (Holloway et al., 2017). Changes in aggregation would have very important impacts on Earth's energy budget independent of anvil cirrus (Bony et al., 2020), and continued study of aggregation may thus prove a doubly beneficial endeavor, as progress would benefit not only our understanding of the ACA feedback but also of Earth's climate sensitivity on the whole.

### 3.2.3 What controls the radiative effect of anvil cirrus on climatological scales?

Even if we were to take for granted that ACA declines with warming, that alone does not tell us whether changes in ACA are a positive, negative, or neutral feedback. Unlike tropical marine low clouds, which have a uniformly negative CRE, anvil cirrus can act both as positive and negative forcing agents depending on their optical thickness and altitude, which vary over the course of their life cycle (Jensen et al., 1994; Hartmann et al., 2001). Fresh, optically thick anvils have a negative CRE, while thinner, aged anvils have a positive CRE. The sign and magnitude of the ACA feedback will therefore depend on how changes in ACA are distributed across the wide spectrum of anvil cirrus observed in the Tropics. A preferential reduction in thick anvil

coverage, for example, may constitute a positive feedback, while a preferential reduction in thin anvil coverage would be a negative feedback. Because nonuniform changes in the area of thick and thin clouds would alter the mean anvil optical depth, the anvil area and optical feedbacks are closely intertwined; for simplicity, we use the term "ACA feedback" to refer to both changes in cloud area and the resulting changes in mean optical depth.

Constraining the ACA feedback has proved difficult, as evidenced by the fact that it was the least certain of all cloud feedbacks in a recent assessment of equilibrium climate sensitivity by Sherwood et al. (2020). In that paper, ACA feedback was estimated to be $-0.2$ W m$^{-2}$ K$^{-1}$ with a likely range of $-0.4$-$0$ W m$^{-2}$ K$^{-1}$. In apparent disagreement with that assessment, it is our opinion that a positive ACA feedback cannot be ruled out at this time. As we discuss below, attempts to constrain the ACA feedback from interannual variability have produced mixed results, and potential changes in anvil optical depth have not been yet been adequately studied.

Attempts to constrain the ACA feedback have so far produced mixed results. Model-based assessments are inconclusive on the whole (Mauritsen and Stevens, 2015; Li et al., 2019; Zelinka et al., 2022) and have relied primarily on GCMs, which, in our opinion, do not reliably simulate anvil processes and parameterize cloud fraction. Assessments relying on observations of interannual climate variability have taken varied methodological approaches and have produced estimates ranging from strongly negative to slightly positive (Ramanathan and Collins, 1991; Lindzen et al., 2001; Lin et al., 2002; Chambers et al., 2002; Williams and Pierrehumbert, 2017; Choi et al., 2017; Ito and Masunaga, 2022; McKim et al., 2023). These assessments are complicated by the fact that variability in observed CRE is related not only to ACA but also to in-situ formed cirrus, low- and mid-level cloud properties, anvil altitude (FAT), and clear-sky radiative fluxes. Moreover, interannual temperature variability, which is primarily El Niño-Southern Oscillation (ENSO)-driven, is associated with a spatial reorganization of convection that is not necessarily representative of long-term warming. While dealing with these factors is not always straightforward, continued empirical assessment may be worthwhile in the absence of robust theory relating ACA and CRE.

Changes in the climatological CRE of anvil cirrus may result not only from changes in ACA, but also from changes in anvil cloud optical depth. Such changes may arise if fresh, optically thick anvils and aged, thin anvils are affected differently by warming. This seems likely, since thick and thin anvils are affected by different processes and have been shown to be affected differently by changes in convective organization (Stein et al., 2017). Anvil optical depth feedback has not, to our knowledge, been systematically assessed, so its importance relative to changes in ACA is unknown. However, observational work generally suggests that the cirrus population in convective regions thins with warming (Liu et al., 2017; Kubar and Jiang, 2019; Höjgård-Olsen et al., 2022), although it is unclear how much of this thinning comes from changes in anvil optical depth as opposed to changes in the relative amounts of anvil and in situ cirrus. If it is indeed anvil-related to some degree, such thinning would have important implications for climatological anvil CRE. For this reason, among others, we believe that the anvil optical depth question should be urgently pursued in future work.

## 4 TTL cirrus

The tropical tropopause layer (TTL; $\simeq$14–19 km) is a region with ubiquitous optically thin cirrus clouds (Fig. 2). The tropical-mean cloud cover in this layer indicated by CALIOP measurements is approximately 30% (compare with Fig. 1), and in situ aircraft measurements with better sensitivity indicate even higher cloud frequencies (Davis et al., 2010). Given the relatively low optical depths of these clouds, their impact on the top-of-the-atmosphere radiative flux is much smaller than for lower-lying convectively-generated anvil cirrus: TTL cirrus exert a small but non-negligible tropical average net top-of-the-atmosphere radiative effect of about 4 W m$^{-2}$ (Haladay and Stephens, 2009; Lee et al., 2009). TTL cirrus radiative heating within the atmosphere is an important component of the TTL thermal budget, generating a tropical average heating of about 0.1 K day$^{-1}$, with peak values of up to 0.5-1 K day$^{-1}$ (Yang et al., 2010; Fu et al., 2018).

### 4.1 What controls TTL cirrus occurrence and microphysical properties?

Much of the interest in the TTL in general, and TTL cirrus in particular, stems from the fact that upward motion across the tropopause as part of the Brewer-Dobson circulation provides the primary source of air to the stratosphere. TTL cirrus regulate the humidity of the stratosphere by freeze drying the air to approximately the saturation mixing ratio corresponding to the cold tropical tropopause temperature. This mechanism provides an indirect climate impact of TTL cirrus since small changes in stratospheric humidity can have important climate (Forster and Shine, 2002; Solomon et al., 2010; Huang et al., 2020) and stratospheric ozone chemistry impacts (Shindell, 2001).

Upward vertical motion and very low temperatures maintain high relative humidity in the TTL, which leads to higher zonal cloud occurrence frequencies than other regions in the atmosphere. The regional distribution of TTL cirrus is controlled by the TTL thermal structure; rapid horizontal motions transport air into cold tropical tropopause regions, such as the western Pacific, where TTL cirrus form (Holton and Gettelman, 2001). The temporal variability of TTL cirrus occurrence is largely controlled by temperature variability driven by atmospheric waves, including gravity, Kelvin, and mixed Rossby-gravity waves (Boehm and Verlinde, 2000; Immler et al., 2008; Kim et al., 2016; Sweeney et al., 2023). TTL cirrus structure and lifetimes will be affected by wind shear, wave-driven temperature variability, and possibly small-scale dynamics driven by cloud radiative heating (Dinh et al., 2010). However, TTL cirrus may typically be too short-lived for cloud-radiative heating to drive thermal instability (Jensen et al., 2011).

TTL cirrus generally form above the primary detrainment level for tropical deep convection ($\simeq$12–13 km, Fig. 2). However, extreme deep convective systems that overshoot their level of neutral buoyancy and extend up to near the tropical tropopause provide the primary source of water vapor for the TTL (Ueyama et al., 2018). Hence, TTL cirrus do tend to occur frequently in convectively active tropical regions. In some cases, TTL cirrus form above deep convective systems (i.e., pileus clouds), presumably driven by convective lifting and cooling (Garrett et al., 2004). However, it is challenging to disentangle the relationships between convection and TTL thermal variability since convective regions in the tropics tend to coincide with cold tropopause regions.

Nucleation of ice crystals under cold TTL conditions can occur via either homogeneous freezing of aqueous aerosols or heterogeneous nucleation on insoluble solid particles. Numerous laboratory experiments have shown that homogeneous freezing at low temperatures requires very large supersaturation with respect to ice (60–100%)(Schneider et al., 2021 and references therein), and substantial supersaturation is prevalent in the TTL (Rollins et al., 2016). In situ measurements showing bursts of numerous small ice crystals forming at high supersaturations provide direct evidence of homogeneous freezing in the TTL (Jensen et al., 2013, 2022). Heterogeneous nucleation at low temperatures is less well understood, but particle types such as mineral dust, crystalline ammonium sulfate, and glassy organic aerosols may serve as INPs in the TTL (Sec. 5.1). If effective INPs are abundant enough, they could increase TTL cirrus occurrence frequency, affect TTL cirrus microphysics, and possibly impact dehydration efficiency (Jensen et al., 2018a). If homogeneous freezing dominates, then TTL cirrus ice crystal number concentration will be primarily controlled by cooling rate. On the other hand, if heterogeneous nucleation dominates, then the ice crystal number concentration will be limited by the abundance of INPs.

## 4.2 How well are TTL cirrus represented in climate models?

Representing TTL cirrus in global climate models presents some of the same challenges as for anvil cirrus, including the mismatch between global-model grid box sizes and cloud-scale structures and physical processes. A particular issue for TTL cirrus is the importance of small-scale, high-frequency gravity waves for TTL cirrus microphysical processes. Stochastic approaches have been developed recently whereby temperature perturbations at each model time step are randomly selected from Laplace distributions based on quasi-Lagrangian super pressure balloon measurements in the lowermost stratosphere (Podglajen et al., 2016; Kärcher and Podglajen, 2019) (Sec. 2.2). Most global models use bulk microphysics schemes to simulate cirrus clouds; bulk microphysics schemes assume a functional form for the ice crystal size distribution. Since these schemes were developed based on measurements in warmer clouds, it is not clear how well they work for cold TTL cirrus. Maloney et al. (2019) recently used a bin microphysics scheme coupled to the CAM5 global model to simulate TTL cirrus and compare with aircraft observations. In contrast to bulk schemes, bin schemes discretize the size distribution into many size bins. This study highlighted the difficulty of properly simulating TTL humidity, particularly its small-scale variability caused by gravity waves, and cloud properties with a relatively coarse-resolution global model. Further, such a detailed, computationally-expensive microphysical scheme is not appropriate for long-duration simulations of a changing climate.

## 4.3 How will TTL cirrus change in the future?

The issue of how TTL cirrus (and, correspondingly, the stratospheric-entry water vapor mixing ratio) will evolve in a changing climate potentially involves a number of processes and feedbacks. TTL cirrus occurrence is tightly coupled to tropopause temperatures, as indicated by the observed covariance between Quasi-Biennial-Oscillation/ENSO-driven interannual variability in tropopause temperature and TTL cirrus occurrence (Davis et al., 2013). They also follow the seasonal cycle in the tropopause temperatures, largely driven by varying strength of the Brewer-Dobson circulation (Li and Thompson, 2013; Fu, 2013; Tseng and Fu, 2017). As noted above, extreme deep convective systems extending into the TTL are the primary source of water

vapor to the region; hence, increases or decreases in future occurrence of overshooting convection would presumably result in corresponding increases or decreases in TTL cirrus occurrence.

Trends in tropopause temperature, TTL humidity, and deep convection occurrence are difficult to discern from the available measurement records. Ladstädter et al. (2023); Scherllin-Pirscher et al. (2021) have used GPS radio occultation measurements to show a positive trend in TTL temperature, but given the large interannual variability, it is not clear whether the relatively short data record indicates a long-term trend. Assessing TTL humidity trends is even more difficult given the extreme temporal variability and limited observational records.

## 5    Key challenges on the model representation of cirrus and aerosol-cirrus interactions

A lack of proper representation of cirrus clouds in global models compromises the fidelity of climate projections (e.g. Sherwood et al., 2020 and Sec. 3.2). The last few decades have seen major improvements in the model representation of cirrus and in the understanding of observed cirrus cloud properties (Sec. 2). Among the areas of research that have received increased attention are the representation of cirrus cloud fraction and ice water content in GCMs and numerical weather prediction models (NWPs). Additionally, there is a growing recognition of the importance of aerosol-cirrus interactions and their impact on climate.

In the past, cirrus cloud representation in models suffered due to the use of thermodynamic equilibrium assumptions, which are more suited for liquid water clouds (Tiedtke, 1993), since vapor pressure adjustment is much faster over liquid water than over ice (Korolev and Mazin, 2003). These assumptions affect both the macroscopic fractional cloud coverage and the microphysics of cloud ice formation, preventing ice supersaturation (i.e., relative humidity exceeding ice saturation) from occurring (Kärcher and Burkhardt, 2008). A prognostic fractional cloud coverage was implemented in one GCM along with an advanced representation of the impact of aerosols in cirrus formation to simulate cirrus micro- and macrophysical properties consistently (Muench and Lohmann, 2020). The consistency between the simulated cirrus cloud cover and ice supersaturated areas is crucial for accurately representing anthropogenic effects on cirrus. To date, many global cloud microphysical schemes employ a parameterization of ice nucleation as an explicit source of ice crystals from the atmospheric aerosol and solve diffusion equations describing the uptake of water vapor on ice crystals allowing for ice supersaturation to occur over multiple time steps.

The formulation of microphysical two-moment schemes predicting both bulk cloud particle number and mass in GCMs enabled modelers to simulate how cirrus respond to changes in the abundance of liquid solution droplets and solid INPs, considered better ice-forming agents. While cirrus properties are relatively insensitive to changes in homogeneous solution droplet freezing, adding INPs leads to a decrease in total cirrus ice crystal number concentrations. Competing ice nucleation appears to occur frequently during the formation of cirrus clouds (DeMott et al., 2003; Froyd et al., 2022). While significant progress has been achieved in studying aerosol-cirrus interactions by coupling cloud and aerosol schemes within global models, parameterized cirrus and aerosol microphysics must be supplemented by parameterizations of the underlying small-scale dynamical forcing of ice supersaturation. As a result, global-model estimates of radiative forcing from aerosol cirrus interactions remain highly uncertain.

In our view, the study of small-scale processes needs to be prioritized to improve our ability to predict future cirrus evolution and associated climate feedbacks more reliably: ice activity and number concentrations of INPs; magnitude and variability of small-scale vertical wind speeds in ice-supersaturated areas; and vapor growth of cirrus ice crystals. These interconnected factors influence the life cycle-integrated radiative impact of cirrus. The list of topics offered here is not all-inclusive, but follows directly from the research themes delineated above.

## 525  5.1    What are the properties of ice-nucleating particles?

In an ice formation event, solution droplets freeze homogeneously once ice supersaturation exceeds 70% in the TTL. INPs become ice-active and thus nucleate ice crystals within a range of ice supersaturations. Typically, efficient INPs such as mineral dust particles require ice supersaturation values exceeding $\simeq$30% to activate. Early forming ice crystals take up supersaturated vapor and thereby change conditions for further ice nucleation. In this way, INPs, if present, weaken or prevent homogeneous
freezing, thus lowering ice crystal number concentrations in cirrus. Mineral dust is widely recognized to be the most important INP in the upper troposphere, well characterized both in terms of ice activity (Atkinson et al., 2013; Ullrich et al., 2017) and number concentrations (Froyd et al., 2022). However, properties of other INP types remain poorly constrained by observations, especially in the TTL (Sec. 4). For example, ice nucleation properties of black carbon (BC) particles depend strongly on the combustion source and whether they have been processed within clouds (Marcolli et al., 2021). Simulating mineral dust and
BC particles in global models also presents a significant challenge, due to uncertainties associated with source specification, transport, and scavenging by clouds. Comparisons between global-model simulations and observations of dust and BC particles have shown significant discrepancies (Zhang et al., 2020; Lian et al., 2022).

    Aerosols in the TTL consist of aqueous, sulfate-containing particles with other (largely unspecified) organic compounds. When present in sufficiently low number concentrations, INPs have minimal impact on cirrus formation, so that crystal numbers
are instead dictated by cooling rates and temperature (Kärcher, 2022). A detailed model study, constrained by observations, suggests that only unusually large number concentrations of mineral dust or ammonium sulfate INPs have a notable effect on TTL cirrus properties and probability of occurrence (Jensen et al., 2018a). Moreover, the lack of information about the TTL INP properties translates into large uncertainties in aerosol-TTL cirrus interactions.

    Very little is known about ice-forming aerosol particles in anvils and the small-scale thermodynamic environment promoting
the transition of anvils into long-lived, large-scale cirrus. INPs may either be entrained from the ambient air surrounding the anvil or left over from detrained convective outflow (Twohy et al., 2017; Sauter et al., 2019). Ice formation within existing cirrus can only take place in regions with very few ice crystals, as vapor deposition on pre-existing ice crystals generally consumes the available vapor, preventing supersaturation high enough to allow ice nucleation (Kärcher et al., 2006). Nevertheless, in situ homogeneous or heterogeneous ice nucleation may play a role in the maintenance of long-lived anvil cirrus (Jensen et al.,
2009; Hartmann et al., 2018; Sokol and Hartmann, 2020).

## 5.2 How well is ice supersaturation and its dynamical forcing represented in models?

Ice supersaturation determines the properties of cirrus clouds. Water vapor injected by deep convection moistens the upper troposphere and produces ice-supersaturated regions. Adiabatic cooling can generate high supersaturation and cirrus ice crystal number concentrations in localized regions. Cirrus ice is removed by sublimation in warmer or drier air, often caused by large-scale subsidence. Another important sink of ice is sedimentation into lower, subsaturated layers of the atmosphere where the ice will sublimate.

Extent and longevity of large-scale ice supersaturated regions in global models depend on the synoptic situation and the model's ability to simulate the upper tropospheric water budget and temperature fields accurately. Occurrence frequencies and seasonal variations of upper tropospheric ice supersaturation are reasonably well reproduced in the ERA-Interim reanalyses over the North Atlantic (Reutter et al., 2020) compared to satellite observations (Lamquin et al., 2012). However, the fine structure and magnitudes of ice supersaturation within such regions are not well reproduced. This deficiency is tied mainly to the representation of cloud-scale dynamical forcing and cloud ice microphysics in high resolution models. Often, poor vertical resolution contributes to the inability to reproduce thin observed ice-supersaturated layers. The spatial distribution of cirrus ice is particularly strongly affected by temperature fluctuations generated by mesoscale gravity waves (GWs) (Kim et al., 2016; Podglajen et al., 2018). Cirrus ice crystals form as a result of GW activity and may be affected by turbulence. Tropical GW activity is closely associated with deep convective systems (Alexander and Pfister, 1995; Corcos et al., 2021). Mean GW-induced updrafts are at least 10 times faster than the slow, large-scale uplift and are therefore the main driver of homogeneous freezing and aerosol-cirrus interactions (Dinh et al., 2016; Kärcher et al., 2022).

Both climate and cloud-resolving models struggle to realistically represent the spectrum of upper tropospheric updrafts (Nugent et al., 2022; Atlas and Bretherton, 2023). The failure to represent the high-frequency variability of vertical wind speeds and associated mesoscale temperature fluctuations caused by GWs prevent us from accurately representing in situ ice cloud formation. Only a combination of high horizontal ($0.5 - 1\,\mathrm{km}$) and high vertical resolution ($100 - 200\,\mathrm{m}$) in the upper troposphere allows models to resolve GWs and tropospheric updraft variability (Kuang and Bretherton, 2004; Kim et al., 2016; Skamarock et al., 2019). These resolutions are for now beyond the reach of GSRMs.

In the tropics, small-scale turbulence is mainly associated with deep convection (Lane et al., 2003; Podglajen et al., 2017; Barber et al., 2018; Atlas and Bretherton, 2023) and in-cloud radiative heating (Gasparini et al., 2019). In the case of anvil cirrus, radiatively driven dissipation rates of turbulent kinetic energy ($>10^{-3}\,\mathrm{m^2\ s^{-3}}$) are much larger than typical upper tropospheric values, based on cloud-resolving model simulations (Hartmann et al., 2018).

The effect of microscale turbulence in cirrus ice formation has not yet been understood. In fact, ice nucleation and growth take place below the viscous-dissipation scale, at length scales $< 10\,\mathrm{mm}$. Yet, in all cloud models, cirrus ice formation is forced by homogeneous moisture and temperature fields with magnitudes of ice supersaturation set by the much larger resolved (model grid) scales. Thus, turbulence, inhomogeneous mixing and molecular diffusion of moisture and heat affect nucleated ice crystal numbers in unknown ways.

## 5.3 What controls cloud ice growth from vapor?

Depositional growth is the primary source of cloud ice mass and determines cirrus ice crystal habits, such as columns, plates, or bullet rosettes. Deposition growth rates are determined by vapor and heat diffusion through the air and attachment processes at the ice surface that control the incorporation of adsorbed molecules into the crystalline lattice (Libbrecht, 2005; Lamb and Verlinde, 2011). In cirrus models, surface kinetic processes are treated based on deposition coefficients. These coefficients generally depend on ambient ice supersaturation, temperature, pressure, and ice crystal size and decline during ice growth, as facets develop (Pokrifka et al., 2020). As depositional growth rates by diffusion of water vapor decrease with increasing particle size, aggregation takes over as the dominant growth mechanism for crystals with maximum dimensions larger than a few $100\,\mu$m (Sölch and Kärcher, 2011).

Microphysical models are almost exclusively based on assuming ice crystals to be perfect spheres characterized by a single constant deposition coefficient. Cirrus models resort to this crude approximation due to the lack of a complete theory of vapor growth encompassing the full cirrus temperature regime and to prevent adding additional tracers to represent different ice crystal habits. While bulk water uptake can be modeled reliably with this approximation, it does not allow the prediction of habits, which importantly modulate ice cloud radiative effects and ice crystal fall speeds.

More research is needed to advance deposition coefficient models at cirrus temperatures. Measurements of ice deposition coefficients need to be extended to cirrus temperatures. Theory development will most likely be based on controlled laboratory experiments. Measurements in large aerosol chambers may provide a link between laboratory and real-world conditions (Lamb et al., 2017). Any progress in this direction will improve estimates of ice crystal fall speeds and aggregation efficiencies and thus the simulation of anvil life cycle (Sec. 3.1.1).

## 6 The way forward

This section explores several strategies and perspectives for improving our understanding of tropical cirrus clouds. We discuss the need for continued development and updating of cirrus cloud parameterizations, particularly with respect to ice nucleation, growth, and micro-scale turbulence. Discrepancies between coarse climate models, basic physical understanding, and observations of aerosol-cirrus interactions and anvil lifecycle is a cause of concern that will be hard to resolve, as long as cloud-scale processes remain parameterized. We therefore emphasize the benefits of high-resolution models, while acknowledging the remaining challenges associated with small-scale parameterized processes, especially microphysics. In addition, we highlight the potential of various flow-following approaches and both in situ and satellite observations to improve the understanding of cirrus processes. Overall, these approaches, coupled with advances in big data analytics and computing power, offer promising avenues for advancing our knowledge of tropical cirrus clouds.

## 6.1 Representation of ice microphysics in models

While increasing resolution helps to improve the representation of cirrus clouds in models, processes at the molecular level, such as ice nucleation and growth, still need to be parameterized. Cirrus parameterizations will therefore need to be continually developed and updated in light of new knowledge about the types and ice-forming properties of INPs that are, or might become, increasingly present in the upper troposphere. If future research identifies a significant impact of turbulent mixing on cirrus ice formation (Sec. 5.2), the impact of micro-scale turbulence on ice nucleation and growth will also need to be considered in parameterizations of cirrus microphysics. Another case of concern is the proper representation of the ice crystal aggregation process and the subsequent generation of ice precipitation (Sölch and Kärcher, 2010).

An interesting recent development in this field are the Lagrangian microphysical schemes coupled to weather and climate models, which promise breakthroughs in the process understanding of cloud and precipitation formation (Jensen and Pfister, 2004; Sölch and Kärcher, 2010; Grabowski et al., 2019; Morrison et al., 2020). The particle-following approach offers several advantages over more traditional Eulerian approaches, e.g. the microphysical evolution of the cloud and the sources of ice crystals become easier to track, while unresolved turbulence can be considered in ice crystal evolution. Such approaches also eliminate artificial numerical diffusion, which cannot be completely avoided in more traditional modeling approaches. On the downside, particle-based microphysical approaches require large computational resources and a good fundamental understanding of the underlying microphysical properties, such as ice particle shapes (Shima et al., 2020). Nevertheless, once available for climate modeling, Lagrangian microphysics could lead to substantial progress in the understanding of cirrus clouds.

A second strategy in improving model representation of cirrus clouds has been the removal of artificial distinctions and thresholds within microphysics schemes. Eight years ago, a *particle properties* paradigm was first proposed by Morrison and Milbrandt (2015) in which frozen hydrometeor categories of ice versus snow versus hail versus graupel were removed in favor of tracking a rime fraction. Wang et al. (2021) have documented the improved performance of the Predicted Particle Properties (P3) scheme relative to a traditional two-moment schemes for deep convective scenes over the U.S. Other artificial thresholds exist in microphysics scheme, most notably the minimum and maximum bounds imposed on ice crystal number concentrations. Indeed, in a global analysis of ice microphysical process rates, Bacer et al. (2021) found that artificial numerical tendencies contribute to a third of the overall process budget.

However, the careful budget analysis of the underlying microphysical source and sink processes is often neglected (refer to Sec. 5.1 for a discussion on nucleation). Modern microphysical schemes allow the prognostic numerical computation of ice crystal number and mass concentrations; however, most often, only the resulting total cloud ice mass, number, or effective radius is analyzed. Tracking process rates and constructing ice microphysical process budgets are potential avenues to interpret model biases in ice water content, ice crystal number concentration, or cirrus cloud lifetime.

## 6.2 High resolution models

The realistic representation of ice cloud microphysics (Morrison et al., 2020), and its coupling with macrophysics (e.g. Dietlicher et al., 2019), radiation, and climate is a persistent challenge in climate modeling. A first step to simplify and partially resolve this challenge is to operate at the cloud-resolving scale, circumventing the need for separate convective ice microphysics. Multiple studies highlight the influence of updrafts on tropical cirrus clouds (e.g., Sullivan et al., 2016; Donner et al., 2016; Barahona et al., 2017), and convection-permitting resolutions will better capture the distribution of vertical velocities. GSRM simulations therefore represent a step forward in the representation of tropical high cloud properties and their evolution.

In addition, GSRMs explicitly resolve deep convection and the development of anvil cirrus. Since convection is the main source of GWs in the tropics, the amount and realism of resolved GWs should increase in such simulations (Stephan et al., 2019). The same is true for waves from orographic sources and perturbations of the large-scale flow field, which are more common in the extratropics. However, while most low-frequency GW activity is simulated by GSRMs, vertical wind speed variances are not yet fully captured at high intrinsic wave frequencies (Podglajen et al., 2020; Atlas and Bretherton, 2023), which are important for in situ cirrus formation. Current GSRMs do not fully capture vertical wind speed variance as observed by balloons (Köhler et al., 2023), at least in part because the current generation of GSRMs lacks the required horizontal and vertical resolution.

The explicit simulation of deep convection allows GSRMs to realistically represent the diurnal cycle of convection (Nugent et al., 2022), the lifetime of deep convective systems, and their anvil cirrus cloud shield (Feng et al., 2023). GSRMs are capable of simulating cirrus initiated by (overshooting) convection within the TTL (Nugent et al., 2022). Despite their increasing realism and explicit transport of water vapor and frozen water mass fluxes in the upper troposphere, the current generation of models exhibit large intermodel spread and significant biases in, for example, frozen water path, cirrus radiative properties, or upper tropospheric vertical velocity distribution (Roh et al., 2021; Nugent et al., 2022; Turbeville et al., 2022).

Large intermodel spread in high resolution simulations is not a surprise, since cloud microphysical processes still need to be parameterized. In fact, most of the intermodel spread in GSRMs can be attributed to the parameterization of cloud microphysics and its coupling to the dynamical forcing (Nugent et al., 2022; Turbeville et al., 2022). This point emphasizes the need for continuous development and updating of ice microphysical schemes and cirrus parameterizations. Unfortunately, the microphysical schemes used in the current generation of GSRMs are often inferior to those used in state-of-the-art GCMs and are therefore inadequate for addressing questions related to, for example, tropical anvil cirrus cloud climate feedbacks. Fortunately, some of these biases can be addressed by relatively simple modifications to the ice microphysical schemes, e.g. by removing unphysical limiters on microphysical process rates (particularly ice crystal number) that are often present in microphysical codes (Atlas et al., 2022).

## 6.3 Evolution perspective

Both observational and modeling data typically provide only snapshots or temporal averages of the current state of clouds in the atmosphere. While these insights help us understand the micro- and macrophysical properties of anvil cirrus, they may not

provide enough detail to understand the underlying processes, such as ice crystal formation. A deeper understanding of such processes would be of particular significance for anvils, whose rapid evolution has substantial effects on both precipitation and radiative fluxes. As a more natural perspective on the evolution of cirrus clouds, Lagrangian studies provide a level of process understanding that is impossible to achieve with Eulerian analysis. Three such approaches have been used to date to advance knowledge of tropical cirrus.

1. Storm Tracking: Storm tracking is valuable for understanding the evolution of anvil cirrus because it helps track the development and movement of the parent storm that is the source of anvil cirrus (Menzel, 2001; Jones et al., 2023). This is a relatively simple approach, often relying on a single 2D variable such as brightness temperature or outgoing longwave radiation obtained from geostationary satellites or model output (Fiolleau and Roca, 2013). Storm tracking provides information on the temporal and size evolution of mesoscale convective systems, thick anvil cirrus cloud shields, as well as microphysical and radiative properties of cirrus clouds (Protopapadaki et al., 2017; Wall et al., 2018; Roca et al., 2017). It can also help explain the response of anvil cirrus properties to interannual climatic variability (Sullivan et al., 2019). However, storm tracking algorithms can only provide information about the developing and mature stages of anvil evolution, as the detection of thinner anvils (below a cloud optical depth of 3) can be biased by signals from lower-lying clouds (Bouniol et al., 2016).

2. Trajectories: Tracking air parcel trajectories can provide valuable information about the processes governing cloud formation, evolution, and decay, and can also improve atmospheric models and the interpretation of airborne measurements or remote sensing data (Sullivan et al., 2022). Tracking detrained clouds and water vapor has provided new insights into the life cycle of tropical cirrus, revealing that their lifetimes are longer than expected from ice crystal sedimentation rates (Luo and Rossow, 2004; Mace et al., 2006). Similar methods have been used to study cirrus cloud microphysical origin in both tropical and extratropical regions (Luo and Rossow, 2004; Wernli et al., 2016). Jensen et al. (2018b) used a Lagrangian particle approach to study the evolution of ice crystals transported by subtropical convection, highlighting the importance of sedimentational size sorting and growth by deposition. Gasparini et al. (2021) used offline trajectories coupled to high-resolution GCM data to study changes in anvil evolution between the present and a warmer climate. Meanwhile, a study using trajectories coupled with a CRM and an offline microphysical scheme highlighted the large uncertainties associated with the description of cirrus microphysical processes (Sullivan et al., 2022). Innovative uses of trajectories coupled with simple microphysical modeling can also significantly enhance the value of airborne and satellite aerosol and cloud measurements, providing important insights into the sources of in situ cirrus clouds, such as dust INPs (Froyd et al., 2022).

3. Passive tracers: While storm tracking and trajectory analysis offer many benefits, they can be challenging to implement. Passive tracers offer an alternative approach, often used in the analysis of stratospheric circulation (Waugh and Hall, 2002) or as a transport diagnostic for climate models (Krol et al., 2018). Though they cannot track individual cloud parcels and can only be used in a model framework, tracers can provide valuable aggregate information about cloud evolution and microphysics. Because atmospheric models generally allow for easy implementation of tracers, they come

with a modest computational cost. In previous studies, cloudy updraft tracers were used to investigate changes in anvil temperature in different climate states (Hartmann et al., 2019) and to study the impact of insolation on cloud lifetime (Gasparini et al., 2022). Tracers can also be adapted to track the time after the onset of a particular microphysical process, such as ice nucleation, making them useful for studying ice crystal evolution pathways. The relative amount of heavier deuterium compared to hydrogen and oxygen-18 compared to oxygen-16 in water is strongly influenced by phase transitions. These ratios can be measured in the atmosphere both by in situ and remote-sensing data (Galewsky et al., 2016 and references therein). Stable water isotopes can therefore be used as real-world tracers of the atmospheric water cycle and ice evolution pathways as they contain information on air parcel's evaporation/sublimation and condensation/deposition history. Isotopes proved particularly useful in studies of the role of deep convective transport in the formation of tropical cirrus of both convective and in situ origin (Blossey et al., 2010; de Vries et al., 2022).

We believe that a more frequent use of the evolution perspective in both model studies or satellite and in situ observations coupled to reanalysis data is a way to increase the knowledge of cirrus cloud processes, especially in the coming era of high-resolution cloud-resolving model simulations.

## 6.4 Airborne and satellite observations

Several aircraft campaigns have recently collected data on tropical cirrus clouds, allowing direct measurements of ice crystal properties (Sec. 2.2). However, most of these measurements have been taken in cirrus clouds located away from convective sources, providing limited information about the microphysics within deep convection or freshly formed anvil cirrus. To better understand how ice crystal properties evolve in aging anvil cirrus, it is crucial to conduct measurements at various stages of their lifecycle, including the tops of deep convective systems.

Moreover, aircraft field campaign data are key to reducing the large uncertainties that remain regarding the role of heterogeneous nucleation in cirrus clouds and the associated aerosol cirrus indirect effects (Sec. 5.1). Despite extensive laboratory studies, relatively few direct measurements of INP have been made in the upper troposphere. We thus believe INP measurements at temperatures colder than -35°C are urgently needed to address the important open questions about the roles of heterogeneous ice nuclei in both TTL cirrus and evolving anvil cirrus.

Satellite observations are valuable for studying tropical cirrus clouds and providing context for in-situ measurements. However, there is a need for further progress in developing satellite products that directly contribute to understanding cirrus formation processes, such as ice crystal number concentration. While such products have recently emerged (Sec. 2.1), they still show several biases that require attention and resolution (Sourdeval et al., 2018; Krämer et al., 2020). It is crucial to improve the representation of particle size distribution (PSD) in satellite remote sensing algorithms, especially for small particles, based on recent in-situ observations (Bartolomé García et al., 2023). Additionally, comparing and evaluating satellite observations with in-situ measurements, either directly or in relation to environmental metrics (e.g. the temperature) is essential. Moreover, long-term unified datasets are a key to detecting trends in cloud properties. While the first evidence of such trends has been detected, the recent decommissioning of the CALIOP lidar creates a gap in thin cloud measurements that may complicate the

creation of long-term datasets with future active instruments (e.g. on board of the EarthCARE satellite mission, Illingworth et al., 2015).

Comparing satellite products with models can be challenging due to inconsistencies in physical assumptions between satellite algorithms and models. In particular, differences in assumptions regarding PSDs or the mass-diameter relationship can lead to important differences that should be characterized. Transparent reporting of these assumptions or using different sets of assumptions to provide an ensemble of retrievals can address this issue. Developing adequate satellite simulators will facilitate efficient comparisons between space-based retrievals and modeling results.

Geostationary radiometers provide valuable temporal resolution that low earth orbiting satellites lack but have limited capabilities for cirrus cloud retrievals. However, recent advancements in neural network approaches applied to geostationary observations and learning from active sensors (e.g., Strandgren et al., 2017) show promise and should be explored further.

Dynamical parameters lack observational constraints from satellites, such as vertical velocities, which is crucial for understanding cirrus formation and accurately estimating convective mass flux. This is particularly relevant for thick anvil cirrus and can help clarify their relationship with convective intensity, precipitation, and circulation as well as expected changes in their optical depth in a warmer world. Fortunately, three planned satellite missions will address this gap by combining satellite-derived microphysical parameters with coincident vertical velocities. The Investigation of Convective Updrafts (INCUS; van den Heever et al., 2022) mission will infer convective mass fluxes using microwave radiometers and radars flying in close proximity, while the Earth Clouds, Aerosols and Radiation Explorer (EarthCARE; Illingworth et al., 2015) and the Atmosphere Observing System (AOS; Braun et al., 2022) missions will both use Doppler radar to derive the vertical velocity of cloud particles.

Other future missions have great potential to significantly improve our understanding of cirrus clouds and driving new product developments. For example, sub-millimeter passive microwave scanning can give independent information on ice mass, particle radius and atmospheric thermodynamic variables, particularly when combined with lidar remote sensing of cloud vertical structure (Jiang et al., 2019). High-resolution lidars will enable more accurate and sensitive cirrus retrievals by directly measuring the lidar ratio (defined as the ratio of extinction to backscatter coefficients), rather than assuming a value. Polarized multi-angle radiometers, like the EPS-SG Multi-Viewing Multi-Channel Multi-Polarization Imaging (3MI; Fougnie et al., 2018) instrument, can provide valuable insights into phase detection and characterization of ice crystal habits. Overall, synergistic approaches are encouraged to ensure consistency and improve our representation and understanding of cirrus radiative effects.

In addition, microwave radiances sensitive to cloud ice are becoming important for weather forecasting, with a potential similar to that of microwave temperature soundings (Geer et al., 2017). The Ice Cloud Imager (Eriksson et al., 2020) aboard MetOP-SG will be the first sensor dedicated to ice observations aboard an operational satellite, offering semi-global coverage on a daily basis. The climate and operational weather forecasting communities are increasingly interested in similar observations, as illustrated by recent investigation in the added value of assimilating lidar and radar observations (Janisková, 2015), which could create synergies and lead to advances in the understanding of tropical cirrus.

## 6.5 Taking advantage of big data and modern computing

Computing power doubles about once every 1.2 years, and NASA alone produces 4 Terabytes of new remotely sensed Earth science data per day (Porterfield, 2021). This outflux of data represents both an opportunity and a challenge. The ease of generating more data is tempting relative to the difficulty of carefully analyzing existing data. We may also sacrifice interpretability at the expense of comprehensiveness (e.g., Beucler et al., 2021; Proske et al., 2022).

We propose a few strategies to make progress on tropical cirrus in harmony with big data. First, *machine learning* (ML) has become pervasive as a tool to extract patterns from large quantities of data. The number of clouds and climate research articles mentioning ML has exploded from almost none in 2012 to 200 publications throughout 2019 (Sullivan and Hoose, 2022). Cirrus emerge from certain ML-based cloud classifications (Kurihana et al., 2022; Zantedeschi et al., 2022). Artificial neural networks have been used for cloud top height and ice water path retrievals (Kox et al., 2014; Strandgren et al., 2017; Amell et al., 2022) and to derive a 3D perspective on tropical cloud systems (Stubenrauch et al., 2023) while explainable ML technique was used for the analysis of retrieved cirrus properties related to meteorological and aerosol conditions (Jeggle et al., 2023). But much room remains to explore ML for tropical cirrus, for example in satellite image analysis of anvil optical depth over time.

*Standardization of datasets and metrics* provides a second opportunity for progress. As an example, the MCS tracking datasets discussed in Sec. 2.1 often employ different thresholds or variables to identify convective systems over time, complicating any intercomparisons. A contrasting example of successful intercomparison comes from Krämer et al. (2016) and Krämer et al. (2020) in which ice water content-temperature spaces were constructed to aggregate in situ data from various field campaigns and identify cirrus origin. Such phase spaces or consistent metrics define a common language for the research community. The OLR-albedo space, originally proposed by Hartmann and Short (1980), is another example, used most recently by Turbeville et al. (2022) to compare GSRM representations of tropical cirrus radiative properties. A final example is the use of distributions of cloud-controlling factors (CCFs), or environmental conditions promoting cloudiness. CCFs were initially suggested as a means of understanding the relative influence of flow and aerosol on low clouds and have been primarily used to understand low cloud feedbacks with warming (Stevens and Brenguier, 2009; Klein et al., 2017). Decompositions of tropical cirrus properties with respect to such CCFs could provide further insight into what drives life cycle or why results about anvil coverage differ.

A third strategy is improved *communication of results*. Computational power has been used disproportionately for running high-resolution model simulations and could instead be directed toward sophisticated visualization or videos that provide a more intuitive grasp on output. Innovations in graphics "allow the emergence of a richer relationship between the visualization and the reader, who is now less of a spectator and more of an explorer" (Hidalgo and Almossawi, Mar 2014). Renderings from groups like the NCAR VisLab and the Climate Visualization Laboratory of the University of Hamburg illustrate how powerful these tools can be to grasp the information in large datasets.

A separate, modeling-specific consideration for tropical cirrus is our approach to complexity. Increasing computational power facilitates models of increasing sophistication. Simpler, less expensive models do not become obsolete but can instead

complement the more sophisticated models. This notion has been termed the *model hierarchy*, the utility and dangers of which have been discussed by Held (2005), Jeevanjee et al. (2017) and Maher et al. (2019).

What could such a model hierarchy look like for tropical cirrus? We suggest that a useful three-rung hierarchy consists of 1) limited-area or long-channel radiative-convective equilibrium simulations; 2) Mock-Walker simulations with a prescribed SST gradient and TTL ascent rate; and 3) GSRMs, if possible coupled with an interactive ocean. The more idealized models with no or prescribed circulation are probably sufficient to estimate the total water budget from various cloud microphysics schemes, whereas the full-complexity models are needed to assess anvil feedbacks. As outlined in Jeevanjee et al. (2017), an array of models is key to determine the robustness of our results: We can turn processes on or off in mechanism denial experiments and test sensitivity of our results to domain size, boundary conditions, and physical assumptions and parameters.

## 6.6 A holistic approach

Big data and the model hierarchy both feed into a third question: Given these powerful tools, should our dominant approach be to seek more detailed understanding at a process level and at the smallest scales? Indeed, as we have argued in Sec. 6.1, the balance of ice microphysical sources and sinks is often overlooked in understanding model biases in cirrus cloud representation. However, given the range of scales involved in tropical cirrus, we argue that it would be a mistake to zoom in overmuch with our new tools. More sophisticated ice microphysics schemes do not always have influence on globally averaged radiation and precipitation fluxes (Proske et al., 2022), and the issue of equifinality implies that the more degrees of freedom our model has, the likelier we are to get a reasonable output for the wrong reason (Mülmenstädt et al., 2020).

We advocate instead for another three strategies that we call cross-tool synthesis, cross-scale studies, and cross-discipline efforts. *Cross-tool synthesis* refers to combinations of or comparisons across the tools outlined in Sec. 2. Satellite simulators allow like-to-like comparison of remote sensing data and model output. Microphysical outputs could especially stand to gain in accuracy from these, for example in assessing simulated ice crystal numbers relative to DARDAR. Fusion remote sensing products, like CCCM, help circumvent issues of spatiotemporal insufficiency or offset. And sampling of GSRM output in a subdomain around the flight track facilitates more meaningful comparison to in situ data (e.g., Kretzschmar et al., 2020).

Then *cross-scale studies* emphasize the need to link work on microscale processes to macroscale phenomena and vice versa. Those in the microphysics community have not always prioritized putting their work into a broader context, such that microphysics can be perceived by other scientific communities as in the weeds. There is, however, evidence that uncertain description of microphysics is decisive for large-scale factors, including cloud-radiative heating rates and equilibrium climate sensitivity (e.g., Zhao et al., 2016; Hu et al., 2021b; Sullivan and Voigt, 2021). Propagating ice microphysical uncertainties to macroscale ones should be a focus to motivate further work on small-scale processes and parameters. Likewise, anvil feedback studies could benefit from consideration of microphysics. How might ice microphysical sources and sinks modulate the anvil cirrus cloud amount, stability iris, or fixed-anvil temperature feedbacks? The lack of consensus in RCEMIP simulations on the ACA feedback strength, as well as theoretical work suggesting lack of robustness in the stability iris hypothesis, highlight a role for small-scale processes in constraining these climate feedbacks (Wing et al., 2020; Jeevanjee, 2022).

Following on more studies across scales, *cross-discipline efforts* acknowledge again the importance of better communicating our results—from those more observationally focused to those more theoretically focused and from those working at process scales to those working at climate scales. Dialogue across these groups is key at a time when our models and observations are more powerful than ever, and several initiatives have already tried to bridge the disciplinary divide (e.g. PIRE project [1], projects initiated by the GEWEX Global Atmospheric System Studies Panel, such as UTCC PROES [2]). We look forward to advances on tropical cirrus over the next years, fueled by big data, the model hierarchy, improved communication, and an urgent need for understanding of cirrus cloud-climate interactions.

*Code and data availability.* The plotting scripts and processed data used to produce Fig 1 is accessible at https://github.com/adambsokol/gasparini_et_al_2023.

*Author contributions.* BG initiated and coordinated the writing of this manuscript and wrote part of the manuscript. SCS, ABS, BK, EJ, DLH all contributed to the writing and editing of the manuscript.

*Competing interests.* The authors declare no competing interests.

*Acknowledgements.* BG acknowledges funding received from the European Union's Horizon 2020 research and innovation programme under the Marie Skłodowska-Curie grant agreement No 101025473. The PIRE project (NSF Grant OISE-1743753) stimulated a lot of discussions on the content summarized in this Opinion. SCS was supported by start-up funds from the University of Arizona. DLH and ABS were supported by NSF Grant AGS-2124496, and ABS by NASA FINESST grant 80NSSC20K1613. We appreciate the insightful reviews by Maximilien Bolot and Aurélien Podglajen and the comment by Claudia Stubenrauch. We thank Odran Sourdeval, Fabian Mahrt, Brett McKim and Maximilien Bolot for their comments on early versions of the manuscript.

---

[1] https://www.pire-cirrus.org
[2] https://www.gewex.org/UTCC-PROES/

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
