# Peer review of "Opinion: Tropical cirrus — From micro-scale processes to climate-scale impacts"

_EGUsphere, 2023_

## Referee Comment (RC1)

**Review of Gasparini et al., submitted to ACP**

Maximilien Bolot

Title: Opinion: Tropical cirrus—From micro-scale processes to climate-scale impacts

Manuscript no: egusphere-2023-1214

Iteration: Initial submission

**General comments:**

The authors review the recent literature on tropical cirrus and expose their perspective as to where the field is going, which challenges lie ahead, and which important questions must be answered to. Observational, theoretical, and modeling aspects are addressed. Both convectively generated anvil cirrus and in situ generated TTL cirrus are discussed. An important aspect emphasized throughout the review is the very wide range of scales over which the underlying physics of cirrus happens, from ice microphysics to cloud macrophysics. In light of such scale disparity, the authors discuss the new opportunities offered by the next generation of global storm-resolving models, but also new challenges that come with operating at the cloud-resolving scale. The impact of new observational capacities expected to become available in the next decade is also discussed.

I generally like and support this review. Given how uncertainties linked to tropical cirrus are pervasive in many aspects of climate projections, I think it is useful at that stage to reflect upon the progress that has been made in the field and outline the directions of research that should be prioritized. This work is also timely. As noted by the authors, we are at a crossroad of traditional approaches and new approaches fueled by rapid increase in computational and observational capacities. Metrics for measuring progress may have to be redefined and challenges of consistency may require in some cases to rebuild from scratch rather than attempt to unify new developments with already existing parameterizations.

Given that this review is basically an opinion piece, it entails a degree of arbitrariness in the presentation of the material that reflects the choice of the authors and that I will not question. I generally agree with the structure of the manuscript and classify my recommendation as minor revision. However, I have some specific and minor comments, outlined below, that I encourage the authors to address.

**Specific comments:**

One point that I wish the authors had featured more prominently is the added value of water isotopologues for cirrus studies. I suggest that the authors add a few sentences to this regard, in the manner they will find appropriate. I know isotopes are mentioned at line 140 but almost in passing. Isotope packages have been embedded in cloud-resolving models and featured in a number of recent studies. Efforts are also underway in Japan to bring isotopes to the NICAM GSRM. This seems relevant for the scope of this review. In parallel, laboratory measurements of isotope fractionation factors have tested the importance of surface kinetic effects for ice growth at cirrus-relevant temperatures, which is relevant to sections of the manuscript where the deposition coefficients for ice growth are discussed.

I suggest that the laboratory work on ice deposition [1] [2] be referenced near line 580. (See corresponding references at the end of this document.) You could mention for instance that the alteration of the preferential partitioning of water isotopologues by non-equilibrium kinetic effects at ice deposition can inform on surface kinetics and deposition coefficients.

Moreover, Section 6.3.3 (Passive tracers) seems a good place to reference isotopic studies with isotope-embedded CRMs. You could for instance present water isotopologues as a kind of tracer of water vapor and ice pathways in the atmosphere, and mention the modeling studies [3] [4].

Unrelated to this, I have another comment on the concept of supersaturation in the TTL. In Section 4, near line 420, you write: "The fundamental reason why cirrus occur so frequently near the tropical tropopause is the prevailing upward motion that drives supersaturation and cloud formation". I think this is misleading, if not erroneous. Earlier TTL work (e.g. Sherwood and Dessler 2020) attempted precisely to explain why there is no widespread vapor saturation at the mean tropopause temperature despite tropical mean upward motion. The consensus, as you correctly state near line 421, involves lateral transport through cold traps, as first formalized by Holton and Gettelman 2001 and further confirmed by subsequent Lagrangian studies. I therefore suggest that you modify your sentence accordingly and emphasize the Lagrangian cold traps instead of vertical motions in the TTL.

**Minor comments:**

Line 1: "Tropical cirrus clouds". I think you should define cirrus here in the abstract for the broader audience: "Tropical cirrus clouds, i.e. high-level ice clouds, play a critical role [...]"

Line 13: Typo?. "understanding" -> "understand"

Line 21: Perhaps add "in situ ice formation outside of convective clouds" to show that you rule out in situ formation within anvils here?

Line 39: I have difficulty parsing the beginning of that sentence. Perhaps remove the comma: "While thin in situ cirrus in the TTL [...]"

Line 82: Typo. "currently currently". Remove extra occurrence.

Line 130: You may also want to mention CR-AVE in 2006, which paved the way for TC4.

Line 142: Typo. "retreival" -> "retrieval"

Line 188: "GSRMs represent deep convection explicitly". I would prefer "GSRMs resolve deep convection explicitly".

Line 247: Typo. "at the top of atmosphereor surface" -> "at the top of atmosphere or surface"

Line 249: You may want to reformulate "Modeling studies show that the ice microphysics scheme can alter top-of-atmosphere outgoing longwave radiation [...]" to show that you place yourself in the realm of models here.

Lines 260-261: Same. Suggestion of reformulation: "But cloud radiative heating, especially in the upper troposphere, has a more direct influence on circulation, and modeling studies show a strong modulation by ice microphysical factors such as the aerosol dependence of the nucleation scheme [...]"

Line 397: Maybe remove "like" and reformulate "This seems a likely possibility [...]"?

Line 418: Besides climate impacts, you may want to mention that changes in stratospheric humidity can also impact stratospheric ozone chemistry [5].

Line 419-420: See specific comments.

Line 425 (near): You may also want to mention the seasonal cycle in TTL cirrus here, that follows the seasonal cycles in tropopause temperature and Brewer-Dobson circulation.

Line 460: Maybe you can mention the seasonal cycle in tropopause temperature again here, besides the temperature variability driven by QBO and ENSO.

Line 476: Perhaps you could add "since vapor pressure adjustment is much faster over liquid water than over ice." after "liquid water clouds" and cite Korolev and Mazin 2003 [6] here. This makes the argument easier to follow.

Line 476 (bis): Typo. Delete comma after "both"

Line 510: Typo. "challenges" -> "challenge"

Line 557: I am not sure I understand the word "surprising" in this context. Did you mean "unsurprising" instead?

Line 561: I think you should remove "the last steps before ice nucleation in cirrus formation" or reformulate as I am not sure what is meant here.

Line 561 (bis): On the particular issue of ice formation in turbulent flows that is raised here, one possibility is to build upon the work on droplet growth in isotropic turbulence (e.g. Lanotte et al 2008 [7]). You could mention that here.

Line 564: You should add "vapor **and heat** diffusion through the air [...]" as the necessity to extract latent heat across the diffusive boundary layer surrounding crystals creates thermal impedance on ice growth.

Line 579: I would also cite Lamb et al 2017 here [2].

Line 680: Typo. "schemes" -> "scheme"

Line 685-693: See my specific comments. This may be a place to mention that water isotopologues have been used as a tracer of the pathways of vapor and moisture and reference modeling studies.

Line 736: I think you should define "lidar ratio" (extinction to backscatter coefficient) here as one cannot assume that every reader will be familiar with this concept.

Line 766: Did you mean "GSRM" instead of "SRM"?

Line 781: Typo. Add space between Held (2005) and Jeevanjee et al. (2017)

**Suggested references:**

[1]  J. Nelson, "Theory of isotopic fractionation on facetted ice crystals," *Atmospheric Chem. Phys.*, vol. 11, no. 22, pp. 11351–11360, 2011, doi: 10.5194/acp-11-11351-2011.

[2]  K. D. Lamb *et al.*, "Laboratory measurements of HDO/H2O isotopic fractionation during ice deposition in simulated cirrus clouds," *Proc. Natl. Acad. Sci. U. S. A.*, vol. 114, no. 22, pp. 5612–5617, May 2017, doi: 10.1073/pnas.1618374114.

[3]  P. N. Blossey, Z. Kuang, and D. M. Romps, "Isotopic composition of water in the tropical tropopause layer in cloud-resolving simulations of an idealized tropical circulation," *J. Geophys. Res. Atmospheres*, vol. 115, no. 24, Dec. 2010, doi: 10.1029/2010JD014554.

[4]  A. J. de Vries, F. Aemisegger, S. Pfahl, and H. Wernli, "Stable water isotope signals in tropical ice clouds in the West African monsoon simulated with a regional convection-permitting model," *Atmospheric Chem. Phys.*, vol. 22, no. 13, pp. 8863–8895, Jul. 2022, doi: 10.5194/acp-22-8863-2022.

[5]  D. T. Shindell, "Climate and ozone response to increased stratospheric water vapor," *Geophys. Res. Lett.*, vol. 28, no. 8, pp. 1551–1554, 2001, doi: 10.1029/1999GL011197.

[6]  A. V. Korolev and I. P. Mazin, "Supersaturation of water vapor in clouds," *J. Atmospheric Sci.*, vol. 60, no. 24, pp. 2957–2974, 2003, doi: 10.1175/1520-0469(2003)060<2957:SOWVIC>2.0.CO;2.

[7]  A. S. Lanotte, A. Seminara, and F. Toschi, "Cloud droplet growth by condensation in homogeneous isotropic turbulence," *J. Atmospheric Sci.*, vol. 66, no. 6, pp. 1685–1697, 2009, doi: 10.1175/2008JAS2864.1.

---

## Author Comment (AC1)

**Response to reviews, Oct 2023**

We thank both reviewers for their constructive comments. We incorporated them in the revised text.  The manuscript has thus the following substantial changes, mainly motivated by the reviewer's comments:

1. We added a new sketch representing the scales (Figure 3)
2. Section 2.2 now includes a short description of balloon measurements
3. We included a short text describing cloud chambers in section 2.3
4. The TTL cirrus section is for consistency with other sections separated into 3 subsections.
5. Isotopes are mentioned in section 6.3
6. We included numerous minor improvements of the text.

Please find specific answers below:
* * *
Referee Maximilien Bolot

1. Stable water isotopes

One point that I wish the authors had featured more prominently is the added value of water isotopologues for cirrus studies. I suggest that the authors add a few sentences to this regard, in the manner they will find appropriate. I know isotopes are mentioned at line 140 but almost in passing. Isotope packages have been embedded in cloud-resolving models and featured in a number of recent studies. Efforts are also underway in Japan to bring isotopes to the NICAM GSRM. This seems relevant for the scope of this review. In parallel, laboratory measurements of isotope fractionation factors have tested the importance of surface kinetic effects for ice growth at cirrus-relevant temperatures, which is relevant to sections of the manuscript where the deposition coefficients for ice growth are discussed. Moreover, Section 6.3.3 (Passive tracers) seems a good place to reference isotopic studies with isotope- embedded CRMs. You could for instance present water isotopologues as a kind of tracer of water vapor and ice pathways in the atmosphere, and mention the modeling studies [3] [4].
Added as suggested some text on isotopes in sections 2.3 and 6.3.

I suggest that the laboratory work on ice deposition [1] [2] be referenced near line 580. (See corresponding references at the end of this document.) You could mention for instance that the alteration of the preferential partitioning of water isotopologues by non-equilibrium kinetic effects at ice deposition can inform on surface kinetics and deposition coefficients.
Added as suggested.

Unrelated to this, I have another comment on the concept of supersaturation in the TTL. In Section 4, near line 420, you write: "The fundamental reason why cirrus occur so frequently near the tropical tropopause is the prevailing upward motion that drives supersaturation and cloud formation". I think this is misleading, if not erroneous. Earlier TTL work (e.g. Sherwood and Dessler 2020) attempted precisely to explain why there is no widespread vapor saturation at the mean tropopause temperature despite tropical mean upward motion. The consensus, as you correctly state near line 421, involves lateral transport through cold traps, as first formalized by Holton and Gettelman 2001 and further confirmed by subsequent

Lagrangian studies. I therefore suggest that you modify your sentence accordingly and emphasize the Lagrangian cold traps instead of vertical motions in the TTL.
Modified to "Upward vertical motion maintains high relative humidity in the TTL, which leads to higher zonal cloud occurrence frequencies than other region in the atmosphere."

We still believe upward motion is the fundamental reason for the TTL's higher relative humidity than other parts of the atmosphere. Nevertheless, as mentioned by the reviewer, the role of horizontal transport is not to be neglected.

**Minor comments**
Line 1: "Tropical cirrus clouds". I think you should define cirrus here in the abstract for the broader audience: "Tropical cirrus clouds, i.e. high-level ice clouds, play a critical role [...]"
Done.

Line 13: Typo?. "understanding" -> "understand"
Kept understanding.

Line 21: Perhaps add "in situ ice formation outside of convective clouds" to show that you rule out in situ formation within anvils here?
Done.

Line 39: I have difficulty parsing the beginning of that sentence. Perhaps remove the comma: "While thin in situ cirrus in the TTL [...]"
Line 82: Typo. "currently currently". Remove extra occurrence.
Line 130: You may also want to mention CR-AVE in 2006, which paved the way for TC4.
Line 142: Typo. "retreival" -> "retrieval"
Line 188: "GSRMs represent deep convection explicitly". I would prefer "GSRMs resolve deep convection explicitly".
Line 247: Typo. "at the top of atmosphereor surface" -> "at the top of atmosphere or surface"
Line 249: You may want to reformulate "Modeling studies show that the ice microphysics scheme can alter top-of-atmosphere outgoing longwave radiation [...]" to show that you place yourself in the realm of models here.
Done/included in the text as suggested.

Lines 260-261: Same. Suggestion of reformulation: "But cloud radiative heating, especially in the upper troposphere, has a more direct influence on circulation, and modeling studies show a strong modulation by ice microphysical factors such as the aerosol dependence of the nucleation scheme [...]"
Kept as is to avoid making a long sentence even longer.

Line 397: Maybe remove "like" and reformulate "This seems a likely possibility [...]"?
Line 418: Besides climate impacts, you may want to mention that changes in stratospheric humidity can also impact stratospheric ozone chemistry [5].
Done.

Line 419-420: See specific comments.
Line 425 (near): You may also want to mention the seasonal cycle in TTL cirrus here, that follows the seasonal cycles in tropopause temperature and Brewer-Dobson circulation.
Line 460: Maybe you can mention the seasonal cycle in tropopause temperature again here, besides the temperature variability driven by QBO and ENSO.
Added a sentence on the seasonal cycle near former line 460 (now line 485).
"They also follow the seasonal cycle in the tropopause temperatures, largely driven by varying strength of the Brewer-Dobson circulation (Li et al., 2013, Fu, 2013, Tseng et al., 2017)."

Line 476: Perhaps you could add "since vapor pressure adjustment is much faster over liquid water than over ice." after "liquid water clouds" and cite Korolev and Mazin 2003 [6] here. This makes the argument easier to follow.
Done.

Line 476 (bis): Typo. Delete comma after "both" Line 510: Typo. "challenges" -> "challenge"
Line 557: I am not sure I understand the word "surprising" in this context. Did you mean "unsurprising" instead?
Line 561: I think you should remove "the last steps before ice nucleation in cirrus formation" or reformulate as I am not sure what is meant here.
Done/included in text.

Line 561 (bis): On the particular issue of ice formation in turbulent flows that is raised here, one possibility is to build upon the work on droplet growth in isotropic turbulence (e.g. Lanotte et al 2008 [7]). You could mention that here.
While it is true that a number of studies addressed the impact of homogeneous turbulence on cloud droplet growth, we do not like to draw a parallel to cirrus ice formation here for a number of reasons. For instance, the stratified turbulence environment in which cirrus ice formation takes place differs fundamentally from that in low-level liquid-phase clouds. Moreover, CCN activation and growth is a reversible process while ice nucleation is not.

Line 564: You should add "vapor **and heat** diffusion through the air [...]" as the necessity to extract latent heat across the diffusive boundary layer surrounding crystals creates thermal impedance on ice growth.
Line 579: I would also cite Lamb et al 2017 here [2]. Line 680: Typo. "schemes" -> "scheme"
Line 685-693: See my specific comments. This may be a place to mention that water isotopologues have been used as a tracer of the pathways of vapor and moisture and reference modeling studies.
Line 736: I think you should define "lidar ratio" (extinction to backscatter coefficient) here as one cannot assume that every reader will be familiar with this concept.
Line 766: Did you mean "GSRM" instead of "SRM"?
Line 781: Typo. Add space between Held (2005) and Jeevanjee et al. (2017)
Done.

Referee Aurelien Podglajen

General Comments:

1) Availability of spaceborne observations:

I don't entirely share the authors' optimism regarding the "growing abundance of observational data.", since not all instruments have equal relevance for monitoring (thin) ice clouds. A significant reduction in available spaceborne observations has occurred since the ending of CALIOP on August 1st (https://www-calipso.larc.nasa.gov/). While future active spaceborne instruments (EarthCARE, etc.) are planned, the lack of overlap between upcoming spaceborne lidars and CALIOP might complicate the creation of long-term datasets necessary for trend analysis in the context of climate change. The authors might wish to mention this type of issue in the paper.
Added a text on it on page 25-26, lines 743-745.

2) Balloon-borne observations

Both spaceborne and in situ aircraft measurements are reviewed, but somehow balloons are mostly omitted from the outlook section (Sect. 6), although a few studies based on data from this platform are cited elsewhere in the paper. Given the emphasis put on cloud tracking, life cycle and Lagrangian modeling techniques (Sect. 6.3-6.4 in particular), I am wondering whether the authors would like to provide a brief account of the possibilities offered by balloon soundings (radiosonde and long-duration balloons). Recently, a renewed effort in the development of lightweight instrumentation has been undertaken and balloons have been deployed in a few of campaigns (only to cite a few studies: Cirisan et al., 2014; Khaykin et al., 2016; Wolf et al., 2018, Ravetta et al., 2020, Kalnajs et al., 2021, Bramberger et al., 2022). Dynamical information inferred from quasi-Lagrangian balloons has also been used in a number of studies of TTL cirrus (Jensen et al., 2016; Corcos et al., 2023).
Added a paragraph on balloon observations in Section 2.2, lines 146-154.

3) Presentation

To help the reader, some terms and concepts could be more clearly defined (e.g.,whether TTL cirrus are solely in situ or can be convectively detrained, what is meant by 'convective origin', 'small scales'). Further details are provided below.

Specific Comments:

- Line 5-6: "a growing abundance of remotely sensed and in situ observations": I'm unsure to share this optimism (refer to main comment 1).
Added a comment on it as mentioned under general comment #1 on lines 743-745.

- Line 21-23: These sentences might be interpreted as considering all TTL cirrus to be in situ formed. Could you clarify?
On the other hand, aged anvils originating from TTL-penetrating deep convection can also be a source of TTL cirrus.

- Line 25: The statement isn't supported by Fig. 2; you might include a reference here (Sassen et al., 2009 ?).
Added Sassen et al., 2009 and Yang et al., 2010 citations.

- Line 122: It would be useful provide a quantitative assessment of available data from aircraft campaigns (e.g., show the track of the flights from the Kramer et al Julia database in a figure, or provide the total number of flight hours)?

Done.

- Line 140: Were measurements of ice crystal isotopic composition indeed conducted during those campaigns? As far as I recall, there were only vapor phase measurements

That's right. Changed from "isotopic composition" to "water vapor isotopic composition".

- Line 147: What is the definition of "cirrus clouds originating from deep convection" in that paper ?

Simply based on IWC only. Thicker clouds are assumed to be of liquid-origin. Such origin is in the tropics related to deep convection. Deleted "originate from deep convection" as such statement is hard to do based exclusively on Krämer et al., 2016 & 2020 data.

- Line 149: In my opinion (and also reading the following sentence), it would be more accurate to say that models bridge the gap between laboratory studies and observations.

Reformulated the sentence.

- Line 163-164: Choose between accommodation coefficient and deposition coefficient.
- Line 172: Perhaps specify "numerical" models.
- Line 189: "removing bias…" – You might consider adding supporting references.
- Line 192-196: Introduce references for these points.

Done.

- Line 198-199: Does this not apply to midlatitude cirrus as well?

That can be true particularly for thicker, homogeneously nucleated cirrus (e.g. Joos et al., 2014), but the range of optical depths and thus radiative effects in midlatitude cirrus is much more limited compared with tropical anvils. Cirrus thick enough to have a substantial cooling impact are thus not that frequent. Moreover, the strong sunlight seasonality complicates the net CRE of  cirrus in midlatitudes.

- Line 241-242: Please consider elaborating on this point further.

*The decay timescale is therefore strongly dependent on temperature (colder temperatures imply slower sublimation, Seeley et al., 2019), environmental relative humidity and size of ice crystals (Jensen et al., 2018), as well as the mesoscale and synoptic-scale dynamical support.*

Anyone would like to add something here (I added the temperature point)? I don't think we have good citations for it.

- Line 258: Suggest replacing 'drive' with 'be related to.'

Done.

- Line 280: An example of such phenomenon is provided by Ferlay et al., 2014.

Added.

- Line 333: Could you specify a range?

Correlations between SST and anvil cloud area are sensitive to the boundaries of the study region, thresholds used for anvil identification, etc. We don't feel that specifying a range of correlation coefficients found in previous work would be particularly useful to the reader at

this point in the paper, but this information is available from the references found in the paragraph.

- Line 358-360: Would you say that the stability Iris mechanism is still disputed or widely accepted ? If a firm opinion emerges from the literature, it would be helpful stating it here.

Our assessment of the Stability Iris hypothesis is as follows:
- Models and observations are broadly consistent with the basic physics of the Stability Iris hypothesis
- There seems to be growing consensus that while Stability Iris successfully describes the first-order response of anvil area to warming, other factors are clearly important.

Because of this nuance, we would like to avoid making a blanket statement of whether or not the hypothesis is widely accepted. We think that the current text of this paragraph reflects our opinion that while Stability Iris is broadly consistent with most available evidence, it does not provide a complete answer to the anvil cloud area question. This assessment is consistent with a recent overview provided by Jeevanjee (2022).

- Lines 391-394: You might wish to expand upon this point.

We have elaborated on this point in the text and moved this paragraph to a point earlier in the section, where we believe it makes more sense.

- Line 400: Maybe refer to the paragraph above or mention again that the cited studies leverage SST variations related to different modes of variability (i.e. their conclusions may not apply to global warming).

Not sure we need to do anything here.

- Line 429: Change "the level of neutral ..." to "their level of neutral…"

Done.

- Line 430: I unsure what is meant here. Is it that the correlation between TTL cirrus and convection is related to the detrainment of water more than to the cold TTL temperature anomalies in convective areas (cold traps)? If yes, please clarify, this may seem contradictory with the idea of a cold trap and a few observational studies (Kim et al. 2016, Randel et al., 2015).

We think the paragraph is clear. Detrainment of ice and water from overshooting convection, pileus formation above convection, and the correlation between convection and cold tropopause temperatures likely all contribute to the correlation between convection and TTL cirrus. We don't know which of these processes is dominant.

- Line 451: I believe there is a typo and the citations 'Podglajen et al., 2017; Karcher and Jensen (2017)' should be replaced by 'Podglajen et al., **2016;** Karcher and Podglajen (2019)'

Done.

- Line 475: It would be good to include a reference here.

Added a reference to Tiedtke, 1993.

- Line 494: I would recommend distinguishing between small scales and micro scales to avoid confusion.

We have included a new figure, which semi-quantitatively shows that small scales encompasses both microscale and eddy-scale processes, those that must be parameterized, even in storm-resolving models.

- Line 500: "sharp threshold" sounds redundant to me.

Done.

- Line 509: Reference(s) for the controversy would be helpful.

The controversy alludes to the large spread in ice-nucleation properties of BC particles from various sources determined mostly in laboratory measurements (see, e.g., Kanji et al., 2017). It appears that the advanced physical understanding of soot-induced ice nucleation laid out in Marcolli et al. (2021) based on the Pore Condensation and Freezing mechanism has the potential to understand and explain these differences.

Instead of adding references, we opt to remove "… and have been a controversial issue in recent years" in L508f as we do not want to expand on a relatively (in the overall context of this Opinion piece) minor aspect.

- Line 530: Change 'ejected' to 'injected.'
- Line 542: maybe change vertical wind to temperature anomaly, since the waves considered in those studies mostly have a signature in temperature (not vertical wind)

Done.

- line 543: I am not sure that the Mace et al., 2006 reference is appropriate here. Maybe Alexander and Pfister (1995)?

Done.

- Line 545: Maybe specify the type of scales which is referred to as 'large scale' – planetary or large-scale equatorial waves.

Changed to synoptic-scale.

- Line 553: On turbulence in the vicinity of convection, you might also cite earlier studies by Lane et al., 2003; Podglajen et al., 2017; Barber et al. 2018
- Line 613-614: I would replace 'sources of artificiality' by 'arbitrary thresholds and limiters' or something along those lines
- Line 615: 'numerical' → 'artificial'

Done.

- Line 617-622: The whole paragraph needs to be rephrased.

We would rather keep the paragraph in its current form.

- Line 653: 'evolution perspective': perhaps consider replacing 'evolution' by 'life cycle'

We prefer to keep "evolution" as it's a bit more of a universal word than life cycle.

- Line 781: 'both'? (there are 3 citations)

Done.

- line 802: Lagrangian sampling of models: I am not quite sure what is meant here. The citation is in any case not appropriate (neither 'Lagrangian' nor 'trajectory' appear in the cited paper)

We have reworded this as "Sampling of GSRM output in a subdomain around the flight track". We refer to taking model output values in a spatiotemporal bubble around the aircraft position to more accurately evaluate the model against observations. This is indeed the methodology used in Kretzschmar et al. 2020. From their Section 2.2: "For the comparison of our ICON simulations to the ACLOUD data, we temporally and spatially collocate the model output to be consistent with the actual position and altitude of the aircraft."

- Line 802: Could you elaborate on the microphysical uncertainty you are referring to ?

Rephrased.

Please make sure that the references are listed in alphabetical order.

In-text ones? They are now organized in chronological order.
* * *
Additional discussion comment by Claudia Stubenrauch

Concerning the last phrase on 'improved communication': Perhaps one could mention the international effort to build working groups like GEWEX UTCC PROES (Process Evaluation Study on Upper Tropospheric Clouds and Convection, https://www.gewex.org/UTCC-PROES/) or other specific projects initiated by the GEWEX GASS (Global Atmospheric System Studies) Panel (https://www.gewex.org/panels/global-atmospheric-system-studies-panel/gass-projects/).

We considered the comment and mentioned two such initiatives in Section 6.6.

**References:**

Joos et al., 2014, doi:10.5194/acp-14-6835-2014
Jeevanjee et al., 2022, doi: 10.1029/2022MS003285
Jensen et al., 2018, doi: 10.1029/2018JD028832
Kretzschmar et al., 2020, doi: 10.5194/acp-20-13145-2020
Tiedtke, 1993, doi: 10.1175/1520-0493(1993)121<3040:ROCILS>2.0.CO;2
Sassen et al., 2009, doi: 10.1029/2009JD011916
Seeley et al., 2019, doi: 10.1029/2018GL080747
Yang et al., 2010, doi: 10.1029/2009JD012393

---

## Referee Report (RR1)

**Review of Gasparini et al., submitted to ACP**

Maximilien Bolot

Title: Opinion: Tropical cirrus—From micro-scale processes to climate-scale impacts

Manuscript no: egusphere-2023-1214

Iteration: Revision

**General comments:**

The revised manuscript has been significantly improved and my concerns have been addressed. I recommend publication of the study after the minor technical fixes listed below.

The only point which can be debated in my opinion is the role of vertical transport in setting final dehydration and the high occurrence frequency of cirrus in the TTL. Multiple studies (Gage et al. 1991, Sherwood 2000, Holton and Gettelman 2001, Hartmann et al. 2001, Fueglistaler et al. 2004) indicate that air masses around the tropopause over the maritime continent, where the coldest temperatures are found, descend rather than ascend. This region plays an important role in determining the water vapor mixing ratio of tropical troposphere-to-stratosphere transport, highlighting the role of horizontal transport through Lagrangian cold traps. However, I do not think that this point is central to the paper and thus do not recommend further modification. Maybe at line 439 you could write: "Upward vertical motion and very low temperatures maintain high relative humidity in the TTL […]"

**Technical corrections:**

Line 149: reel down -> reel-down

Line 720: "Isotopes proved particularly useful in studies of the role of deep convective transport, formation of tropical cirrus of both convective and in situ origin". Something seems wrong with the sentence. Did you mean: "[…] studies of the role of deep convective transport in the formation of tropical cirrus of both convective and in situ origin"?

Line 743: "While first such trends have been already detected, […]". I'm not sure about the English. Perhaps "While the first evidence of such trends has been detected, […]" would be clearer?

---

## Author Response (AR2)

We thank both reviewers for their last comments. We considered all of the suggested last edits.
* * *
**Referee Aurelien Podglajen**

Report 1:
I thank the authors for taking my comments into consideration and recommend the paper be accepted for publication. I only have a few further, very minor suggestions.

line 146: "Quasi-lagrangian super-pressure balloons are an additional platform" -> "Weather balloons and quasi-Lagrangian super-pressure balloons are additional platforms" since the first 3 papers cited use weather balloons.

line 471: "semi-Lagrangian" -> "quasi-Lagrangian"

line 639: 'building upon...., the budget analysis .... is often neglected': Don't you mean 'except for the study by and a few others (?) ..., the budget analysis .... is often neglected' ?

Thanks for carefully looking at the text again, we considered all suggestions.
* * *
**Referee Maximilien Bolot**

Report 2:
General comments:
The revised manuscript has been significantly improved and my concerns have been addressed. I recommend publication of the study after the minor technical fixes listed below. The only point which can be debated in my opinion is the role of vertical transport in setting final dehydration and the high occurrence frequency of cirrus in the TTL. Multiple studies (Gage et al. 1991,Sherwood 2000, Holton and Gettelman 2001, Hartmann et al. 2001, Fueglistaler et al. 2004) indicate that air masses around the tropopause over the maritime continent, where the coldest temperatures
are found, descend rather than ascend. This region plays an important role in determining the water vapor mixing ratio of tropical troposphere-to-stratosphere transport, highlighting the role of horizontal transport through Lagrangian cold traps. However, I do not think that this point is central to the paper and thus do not recommend further modification. Maybe at line 439 you could write: "Upward vertical motion and very low temperatures maintain high relative humidity in the TTL [...]"

Technical corrections:
Line 149: reel down -> reel-down
Line 720: "Isotopes proved particularly useful in studies of the role of deep convective transport,
formation of tropical cirrus of both convective and in situ origin". Something seems wrong with the
sentence. Did you mean: "[...] studies of the role of deep convective transport in the formation of

tropical cirrus of both convective and in situ origin"?

Line 743: "While first such trends have been already detected, [...]". I'm not sure about the English.

Perhaps "While the first evidence of such trends has been detected, [...]" would be clearer

Thanks for carefully looking at the text again, we considered all suggestions.